# Tropical-leaning Atlantic Oscillation favors more typhoons toward Asian high-latitude cities

Zeming Wu [1,2], Chundi Hu [1,2,3] ✉, Wenju Cai [4,5,6,7], Chengyang Zhang[8], Tao Lian [3,9], Gang Huang [10,11], Renguang Wu [3,12], Lifei Lin [1] & Dake Chen[3,9]

Poleward migration of Northwest Pacific typhoons brings severe impacts on East Asian high-latitude cities, yet early typhoon climate prediction remains a long-standing scientific challenge. Here we reveal a seemingly-familiar-yet-strange climate oscillation phenomenon, which we name Tropical-leaning Atlantic Oscillation (TAO). Statistical results show that springtime TAO can explain 56% of the variance in a dominant dipole mode of typhoon track variations during July–September of 1979–2023, suggesting that it possesses a robust predictive skill of peak-season typhoon tracks four months in advance. Specifically, springtime TAO is characterized by a sea-level pressure seesaw between the tropical North Atlantic and the Hudson Bay-Davis Strait, relating to the meridional shift of North America-Atlantic subtropical jet stream. It generates cross-seasonal North Atlantic-and-Pacific surface seawater temperature anomalies, thereby triggering Northwest Pacific cyclonic steering flows that tempt (obstruct) typhoons toward East Asian high-latitude (low-latitude) cities during July–September. Climate models project an increasing frequency of positive TAO events. This may potentially contribute to a poleward migration of typhoon activity toward East Asian high-latitude cities as climate warms, yet uncertainty remains due to model biases in simulating tropical surface seawater temperature patterns. Our results highlight an overlooked impact of an emerging internal climatic oscillation on the enhancing typhoon risks toward high-latitudes.

Typhoons (also called hurricanes) rank among the most destructive natural disasters[1], often cause great economic losses and mortality[2,3]. As the climate warms, typhoon activity is becoming stronger and more frequent at higher latitudes than in the past[4–7]. The poleward migrations of typhoons pose escalating threats to coastal communities at higher latitudes, particularly under the anticipated prevalence of future climate scenarios[5,8]. Therefore, assessing the climate change impacts on nearshore typhoon activity is becoming an important research arena[9,10].

Many studies have focused on the responses of typhoon activity to externally forced climate change, with particular focus on variations in surface seawater temperature (SST)[4,11], subsurface ocean heat[4,11], sea ice[12], or warming Tibetan Plateau[13]. Impacts of North Pacific and Atlantic climate shifts on typhoon activity are also highlighted. For example, interdecadal changes in Northwest Pacific (WNP) tropical cyclone (TC) activity can be attributed to the Atlantic Multidecadal Oscillation (AMO), Interdecadal Pacific Oscillation (IPO) and/or Pacific Decadal Oscillation (PDO)[14–20]. While the impacts of shifting internal atmospheric climatic oscillations on changes in typhoon activity are often overlooked. Projections of shifting climate oscillation impacts on poleward expansion of typhoon threats are therefore inherently uncertain[6,8]. However, policy-makers expect the quantification of the

potential impacts and predictive ability ahead of typhoon season to operationally plan management options.

Poleward migration of WNP TC exhibits prominent seasonality, which is related to the distinct responses of late-season and peak-season TC occurrence to a stronger Pacific Walker Circulation[21]. Although the recent WNP typhoon migration trends during peak season (July–September)[21] are well attributed to the warmer subtropical SST due to the increasing greenhouse-gas emissions[4] and Pacific mega-El Niño-Southern Oscillation (ENSO) forcing[22], the WNP typhoon activity may be responding more strongly to the changes in the prominent internal natural climate oscillations, such as the North Pacific Oscillation (NPO)[23,24], North Atlantic Oscillation (NAO)[25–27], and Northern Annular Mode (NAM)[25,26]. Because this climate variability tends to influence the evolution of tropical and extratropical SST modes across seasons, which in turn modulate the later atmospheric circulations, and thereby indirectly affect typhoon activity.

Here we find that the poleward migration mode of East Asian typhoon activity during peak season is strongly connected to a seemingly-familiar-yet-strange climate oscillation in preceding spring (March–May), which shows a pressure seesaw between the low-latitude North Atlantic and the high-latitude Hudson Bay-Davis Strait. The southern part of this oscillation is located at the tropical North Atlantic, considerably distinct from the classical extratropical NAO (Supplementary Fig. 1a, b), thus, we call it the Tropical-leaning Atlantic Oscillation (TAO). There are several points that need to be emphasized: First, due to the great fame and widespread climate impacts of the classical NAO, it stealthily shields other tropical–extra-tropical atmospheric oscillations. In addition, the TAO appears only during cold seasons (Supplementary Fig. 2) when the NAO is most robust, but substantially weakens during warm seasons when the NAO is relatively weaker. As shown in Fig. 1a, once the NAO signal was removed, the cross-latitudinal correlation of sea level pressure (SLP) can directly capture the previously-overlooked TAO signal. These may be the reasons why the TAO was not identified earlier.

It is well known that the NAO mode often expands to the Mediterranean Sea in the Eastern Hemisphere (Supplementary Fig. 1b). In contrast, the southern part of TAO can expand to deep tropics and the whole TAO pattern is located in the West Hemisphere, closer to the tropical Pacific (Fig. 1b). Just as this prominent spatial feature, TAO is more capable of inducing tropical SST climate anomalies, and thereby can significantly modulate typhoon activity in the following summer. The following results will reveal that the TAO mode can provide a perspective for an in-depth exploration of the poleward migration of East Asian typhoons, and also provide insights into the understanding of cross-ocean interactions.

## Results

### Identifying the spring TAO mode
After removing the extratropical NAO signal from the spring SLP anomalous fields, there are significant negative correlations between 0°–40°N and 50°N–70°N (Fig. 1a; NAO signals removed via linear regression) during 1979–2023. This suggests the existence of a meridional atmospheric oscillation in addition to the NAO over the North Atlantic in spring. The position of this oscillation (red circle in Fig. 1a) is clearly different from that of the NAO (blue circle in Fig. 1a and Supplementary Fig. 1a), and it is more oriented towards the tropical region (lower gray diagonal), which is why we refer to it as the TAO. We use region-averaged SLP to define this TAO mode (see "Methods" and red rectangles in Fig. 1b). The SLP pattern of TAO is also closer to the tropics compared to the extratropical NAO (Fig. 1b versus Supplementary Fig. 1b), especially the southern region south of 30°N. The TAO can be straightforwardly identified as the leading empirical orthogonal function (EOF) mode of spring SLP anomalies over the North America–Atlantic region (25°W–105°W, 10°–80°N), which explains a larger proportion of SLP variances (21.4%, calculated as $R^2$ in

the plotted regions, area-weighted; Fig. 1b and Supplementary Fig. 1b) compared with that of the extratropical NAO (16.9%) during spring over the North America-North Atlantic-Western Europe.

The spring TAO is closely linked to the North America-Atlantic subtropical jet shift. A positive phase of the TAO corresponds to a northward shift of the jet exit region (Fig. 1c). Moreover, the TAO index has a robust Pearson correlation coefficient up to 0.85 ($p < 0.01$) with the North America-Atlantic jet meridional shift index (NAJS; see "Methods", Fig. 1c and Supplementary Fig. 3). Further analysis indicates that the TAO is mainly induced by the meridional shift of the North America-Atlantic jet stream. It is well known that divergence tends to occur to the north of the jet axis at the exit region of an upper-level jet, while convergence prevails to the south[27,28]. As expected, a positive phase of the TAO corresponds to a northward shift of the upper-level jet in the exit region (Fig. 1c), which favors the convergence at 300–100 hPa between 10°–35°N, forcing air to descend into the lower troposphere and inducing high pressure in the subtropical surface (Fig. 1d). Meanwhile, divergence appears at 300–100 hPa between 35°–60°N, leading to low-level air being drawn upward and forming a low-pressure system at higher latitudes surface (Fig. 1d). This process generates a meridional pressure dipole characteristic of the TAO.

The wave activity flux (WAF, see "Methods") shows propagation to both meridional sides of the jet stream (Fig. 1e), with wave energy divergence in the mid-to-upper troposphere (Fig. 1f), indicating that the emergence of the TAO originates from jet variability. Additionally, upward-propagating WAF is evident in lower levels around 30°–40°N (Fig. 1f). This suggests that the ocean surface plays a role in TAO pattern, therefore, TAO is also partly associated with the local air-sea coupling. Whereas the NAO-related anomalies are mainly located in higher latitudes (Supplementary Fig. 1c, d) and NAO is influenced by local air-sea interactions and polar vortex[29,30] (Supplementary Fig. 1e, f). Besides, a comparison with other known decadal modes (e.g., AMO and PDO) via the wavelet coherence spectral analysis shows that the TAO is indeed distinct from the NAO (Supplementary Fig. 4), further solidifying its uniqueness.

### Spring TAO takes typhoon activities toward Asian high-latitude cities
Since the TAO is located closer to the tropics, we speculate that it may cause more significant tropical climate anomalies and thus be more significantly related to the WNP typhoons. Next, we consider the relationship between TAO and WNP typhoon activity. For the period 1979–2023, the TAO shows considerable potential in predicting the WNP typhoon activity toward East Asian high-latitude cities in peak season (Fig. 2a). As shown in Fig. 2b, the anomalous typhoon track dipole (TTD), identified as the leading EOF mode of Asia-Pacific typhoon track density (Supplementary Fig. 5a), can be effectively captured by preceding spring TAO. Of note is that the three extreme El Niño years (1982, 1997 and 2015) are excluded in this study, thus, here the TAO-TTD relationship is almost independent of ENSO to a large extent. Given that the relationship between TAO and ENSO is not significant (even in a longer period of 1958–2023; Supplementary Fig. 6a–e), we have tested and found that even in the case of excluding all ENSO years, the TAO–TTD relationship still keeps robust ($R = 0.72$, $p < 0.01$; Supplementary Fig. 6f, g). Accordingly, the cross-seasonal TAO-TTD linkages are generally independent of ENSO signals.

Besides, the TAO pattern can be also obtained by regressing the spring SLP anomalies onto the TTD index (Supplementary Fig. 5b). As such, the TAO index exhibits a strong correlation coefficient ($R = 0.75$, $p < 0.01$) with TTD index, substantially surpassing that of the extratropical NAO ($R = 0.38$; Fig. 2a). The spatial-correlation between the TAO-related typhoon track density and that of TTD reaches 0.94 ($p < 0.01$; Fig. 2b). Similar results can be confirmed with various TC best-track data (Supplementary Table 1 and Supplementary Fig. 7), suggesting that this cross-seasonal linkage from spring TAO to

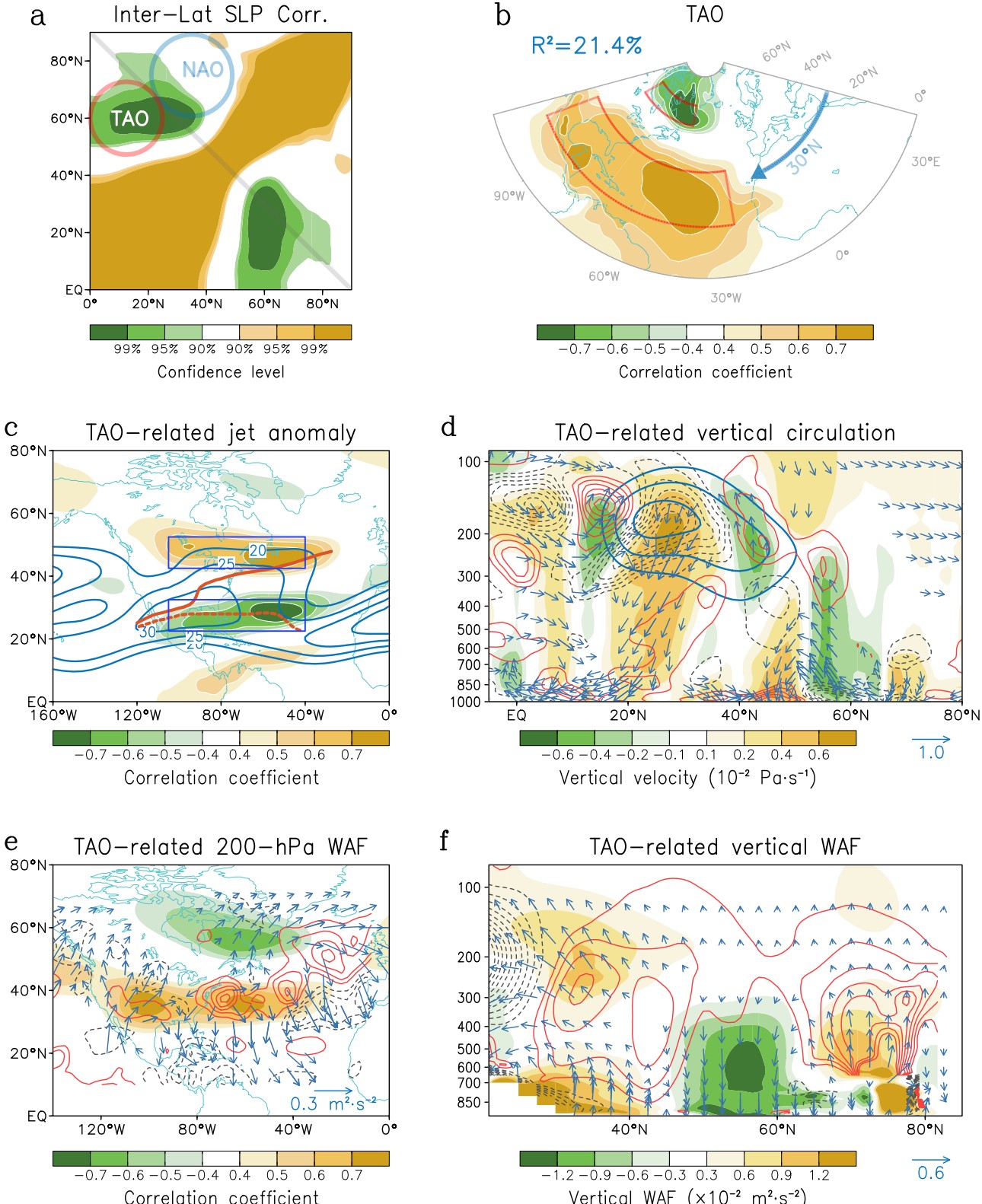

summer TTD is very robust and thereby can be used for seasonal forecasts of typhoon activities in East Asian cities. Of note is that NAJS exhibits a weaker correlation with TTD ($R = 0.64$), albeit NAJS and TAO are highly correlated ($R = 0.85$; Supplementary Fig. 3c). This may be due to that the NAJS index is defined from 200-hPa zonal wind anomalies within a smaller region of the upper-tropospheric jet stream. In contrast, TAO is a lower-tropospheric SLP mode that spans a broader tropical domain, the associated air-sea interactions are

inevitably stronger than that for NAJS, leading to a closer linkage from TAO to TTD.

As shown in Fig. 2c, positive TAO can predict the above-normal typhoon occurrence frequency (measured by track density) and destructive powers (measured by Accumulated Cyclone Energy[31], ACE, see "Methods") of typhoons 4 months in advance, for East Asian high-latitude cities such as Tokyo, Osaka, Kagoshima, Seoul, Qingdao, and Shanghai; meanwhile, the below-normal destructive powers of

**Fig. 1 | Spatial pattern of Tropical-leaning Atlantic Oscillation (TAO) and its dynamical features. a** Inter-latitude correlation between zonally-averaged sea level pressure (SLP) anomalies within 25°W–105°W during 1979–2023. Red and blue circles denote the regions associated with TAO and North Atlantic Oscillation (NAO), respectively. Note that the NAO index is removed via linear regression. **b** Pearson correlation patterns of spring SLP onto the simultaneous TAO index. Rectangles are the regions defining TAO index by SLP differences between (25°W–105°W, 15°N–35°N) and (45°W–95°W, 55°N–65°N). The percentage indicates the area-weighted explained variance in the plotted region. **c** Correlation pattern of 200-hPa zonal wind (U-wind) onto TAO index. Rectangles are the regions defining jet meridional shift index. Blue contours are the climatology of 200-hPa zonal wind

speed (m s$^{-1}$). The thick red solid (dashed) line show the jet axis in positive (negative) jet index year. **d** Vertical cell (vector), vertical velocity (shading) and horizontal divergence (red/black contours for positive/negative; interval by $0.5 \times 10^{-7}$ s$^{-1}$) averaged within 40°W–90°W. Blue contours are the climatology of jet stream (interval by 4 m s$^{-1}$, start from 20 m s$^{-1}$). **e** Wave activity flux (WAF; vector) and its divergence (red/black contours for positive/negative; interval by $1 \times 10^{-7}$ m s$^{-2}$) at 200-hPa. Shading is the correlation pattern of geopotential height onto the TAO index. **f** Vertical cell of WAF (vector), vertical component of WAF (shading) and horizontal divergence of WAF (red solid lines indicate positive, blue dashed lines indicate negative; interval by $0.5 \times 10^{-7}$ m s$^{-2}$) averaged within 40°W–90°W.

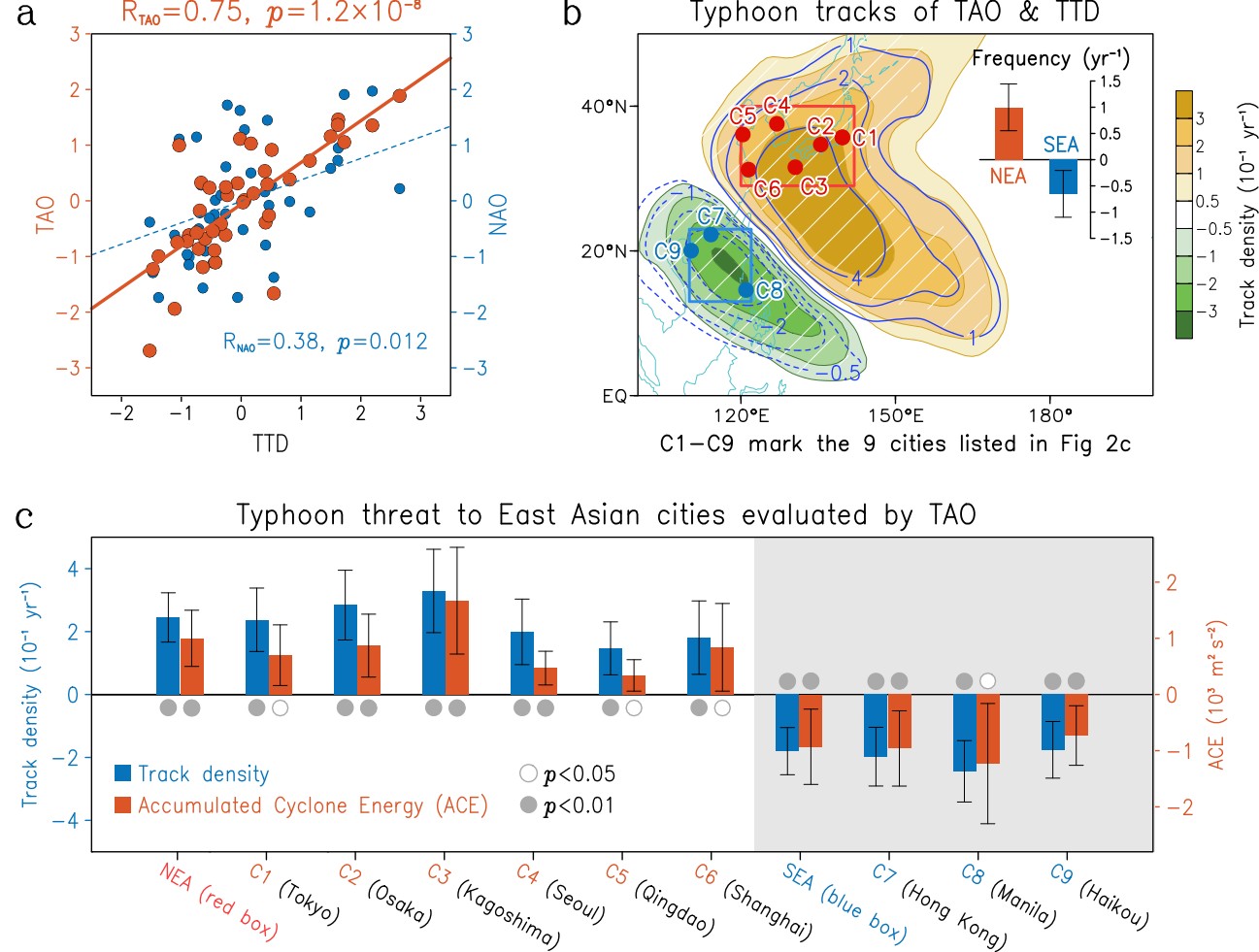

**Fig. 2 | Impacts of spring Tropical-leaning Atlantic Oscillation (TAO) on summer Asia–Pacific typhoon activity. a** Scatter diagram of the spring TAO (red) and extratropical North Atlantic Oscillation (NAO; blue) against the typhoon track dipole (TTD) of peak season typhoon track density during 1979–2023. Correlation coefficient (*R*) between the TAO/NAO and TTD is marked. Straight lines are linear least-squares fits. **b** Regressions of peak season typhoon track density onto the TAO (shading) and TTD (contours) indices. Bars in the upper-right corner are the regression of typhoon frequency entering northern East Asia (NEA, red box) and

southern East Asia (SEA, blue box) onto the TAO index, with the error bar showing the 95% confidence level. **c** Regressions of peak season local track density (blue) and Accumulated Cyclone Energy (ACE; red) onto the spring TAO index for the NEA and SEA regions (marked in **b** with red and blue boxes, respectively) and nine major cities (marked in **b**). The error bar is the 95% confidence interval. Solid (open) circle indicates passing the 99% (95%) confidence level. Three extreme El Niño years (1982, 1997, and 2015) are excluded.

typhoons for East Asian low-latitude cities like Hong Kong, Manila and Haikou can also be foreseen. We further measure the typhoon frequency in each box in response to TAO anomaly (see bar in Fig. 2b). The typhoon frequency is calculated by counting typhoon in peak season entering the boxes in Fig. 2b. A one-standard-deviation change in the TAO index corresponds to an increase of $1.0 \pm 0.45$ yr$^{-1}$ in

typhoon frequency in northern East Asia (see red box) and a decrease of $0.65 \pm 0.45$ yr$^{-1}$ in typhoon frequency in southern East Asia (see blue box). When normalized by box-area, the typhoon-frequency changes are similar in magnitude between the two regions, with about changes of $0.46$ yr$^{-1}$ and $0.41$ yr$^{-1}$ in typhoon frequency per million square kilometers in the southern and northern boxes, respectively.

### Mechanisms of cross-season and cross-basin TAO impacts

Mechanisms of TAO performance in predicting TTD-like typhoon activity can be explained as follows. On the one hand, the springtime TAO can indirectly promote the subsequent Pacific Meridional Mode (PMM) SST anomalies through the TAO-related anticyclonic flows (Supplementary Fig. 8a, b). Since this anticyclonic flows can weaken local trade winds in the subtropical Northeast Pacific (red rectangle in Supplementary Figs. 9 and 11a), and thereby the resultant warm SST, mainly induced by wind–evaporation–SST feedback mechanism, facilitates PMM's development via the seasonal footprint mechanism[32,33].

On the other hand, the TAO can induce the North Atlantic Tripole (NAT) SST anomalies (Supplementary Fig. 8a). Specially, the TAO-related anomalous northeasterly winds over the tropical North Atlantic (TNA) can enhance the trade winds to cool SST there (blue rectangle in Supplementary Fig. 9). This cold SST can further strengthen the anticyclone over the eastern Pacific–North Atlantic via the Gill-type Rossby wave response[34,35]. Therefore, this cold SST can persist into the following summer (Supplementary Fig. 8c) and further contribute to the development of PMM (Supplementary Fig. 8d) through a Gill-type Rossby wave response with an anticyclone to its west[33,36–38]. And the physical process from TNA SST to PMM has been demonstrated by Ham et al.[39] through observations and numerical experiments. However, the impact of TAO on TTD is much more pronounced than that of TNA SST (Supplementary Fig. 10), where only the northward-moving typhoon is significantly altered[40,41]. This may be because that TAO not only affects Pacific climate through the persistent TNA SST anomalies in later seasons (Supplementary Fig. 10c, d), but also directly triggers the formation of the early prototype of PMM via the southern part anticyclonic circulation of TAO during spring (Supplementary Fig. 8a and Supplementary Fig. 11a).

Due to the TAO-promoted summer tropical SST changes, a significant atmospheric response can be induced over the WNP, especially the lower-level WNP cyclonic circulation anomaly (streamlines in Fig. 3a). Moreover, positive PMM linked to TAO can promote active convection over the subtropical North Pacific (Supplementary Fig. 11), inducing a Rossby wave cyclone over the WNP. The lower-level WNP cyclonic circulation can also be identified as the WNP branch of the Pacific–Atlantic subtropical oscillation (PASO)[42,43], driven by the summer TNA SST anomalies via inducing subtropical Walker-cell-like changes (Supplementary Fig. 12) and Pacific convection response (Supplementary Fig. 11b, d), which further contribute to WNP cyclone[42,43]. Correspondingly, there are coherent eastward shifts of enhanced WNP monsoon trough (MT), weakened tropical upper-tropospheric trough (TUTT) and strengthened South Asian high (Fig. 3a). Collectively, these processes favor the eastward shift of typhoon genesis (Fig. 3b) and reinforce the TTD mode of Asia-Pacific typhoon track (Figs. 3c and 2b). Here, we further confirm the relationship between TAO-related SST anomalies and the zonal shift of TUTT and MT. The NAT and PMM SST patterns can partially contribute to the longitude changes in TUTT and MT since their correlations are significant (Supplementary Fig. 13).

Since the strengthened and eastward South Asia high can induce strong descending motions to suppress typhoon genesis in the western part of the WNP[44]. The eastward shift of the WNP MT and the TUTT can drive the favorable conditions for typhoon genesis eastward via changing the typhoon genesis conditions[45]. These environmental modulations on typhoon genesis can be depicted by the Genesis Potential Index (GPI)[46], which shows a zonal dipole (shading in Fig. 3b), suggesting more typhoon genesis over the central-eastern part of the WNP and fewer typhoon genesis over the southwestern part of the WNP (contour in Fig. 3b). To show the relative importance of thermodynamic and dynamic factors in the GPI analysis, a linear decomposition is applied (see "Methods"). The contributions of each variable are calculated, then the regional-mean is obtained for eastern and western regions (outlined in Fig. 3b) associated with positive and

negative GPI anomalies, respectively (Supplementary Fig. 14). The increased GPI in the eastern WNP is mainly contributed by increase mid-level relative humidity, followed by increased lower-level vorticity, ascending motion and increased potential intensity (Fig. 3d). While the vertical wind shear does not significantly contribute to GPI changes. In contrast, the decreased GPI in the western WNP is dominantly contributed by increased vertical wind shear (Fig. 3e).

Furthermore, the typhoon track after genesis is shaped by the steering effects of atmospheric circulation (Fig. 3c). During positive TAO years, a strong anomalous cyclonic circulation dominates East Asia–Pacific during summer, characterized by easterly steering flows in the north and westerly steering flows in the south (vectors in Fig. 3c). High-latitude easterly steering flows are favorable for northwestward movement of typhoons, while low-latitude westerly steering flows prevent typhoon from traveling west. Consequently, more (fewer) typhoons move toward East Asian high-latitude (low-latitude) cities during peak season. And we also quantified the zonal steering flow changes (Fig. 3f), given that the meridional component of steering flow change is negligible. Both northern and southern regions (outlined in Fig. 3c) show significant changes, suggesting important guiding effects for typhoon movement. We further examined the above results after removing the linear trend and found that the relationship between TAO and TTD is still robust, and the related atmospheric and oceanic processes are also significant (Supplementary Figs. 15 and 16), indicating that the TAO mode and its climatic influence are not a signal of global warming trend, but internal climate variability.

### TAO dominates the prediction for East Asian typhoon activities

Given that the spring TAO is a good simulator of peak season typhoon activity, we construct a simple model to predict the TTD time series (Fig. 4a) via multiple linear regression with some other potential prediction factors[23–26,47,48] (see "Methods"). The regression equation derived from the training period of 1979–2014 is as follows:

$$Prediction_{TTD} = 0.682 \times TAO + 0.268 \times NAO - 0.257 \times NAM + 0.086 \times NPO - 0.061 \times SIC - 0.019.$$

The Pearson correlation between observed and predicted TTD is 0.74 for the training period (1979–2014) and even reaches 0.89 for the validation period (2016–2023) (Fig. 4a), both far exceeding the 99% confidence level. The corresponding root mean square error (RMSE) is 0.63 for the training period, and only 0.54 for the validation period. The pahse-sign-agreement is 83.3% for the training period and surprisingly reaches 100% for the validation period (Fig. 4a). This simple model exhibits excellent performance in predicting the TTD index, which can well predict the typhoon anomalies in East Asian coastal cities (Fig. 4b). It should be emphasized that the coefficient of the TAO in the regression formula is notably larger than those of other predictors, suggesting a dominant role of the TAO among the predictors. To further validate the stability of the TAO-TTD relationship and the prominent importance of TAO relative to other factors, we employed the bootstrap method to assess the error range of regression coefficients and used the sliding regression to test the temporal stability of coefficients (Supplementary Fig. 17). This prediction is generally independent of ENSO (Supplementary Fig. 18). Results indicate that only the TAO coefficient in the regression model is statistically significant. Although the relationship between TAO and TTD exhibits some fluctuations, it remains statistically significant throughout the study period.

We further compare the correlation pattern of $Prediction_{TTD}$ (Fig. 4b) with other prediction models established only using TAO (Fig. 4c, d) and NAO (Fig. 4e and Supplementary Fig. 19). The $Prediction_{TTD}$ explains 61.7% of the observed variances in TTD variations during 1979–2023 (evaluated by the $R^2$ between indices; Fig. 4b), while the TAO alone explains a substantial 56.3% of the variation (Fig. 4c). When the NAO signal is removed, TAO maintains its strong linkage to the TTD pattern (Fig. 4d). In contrast, the extratropical NAO-

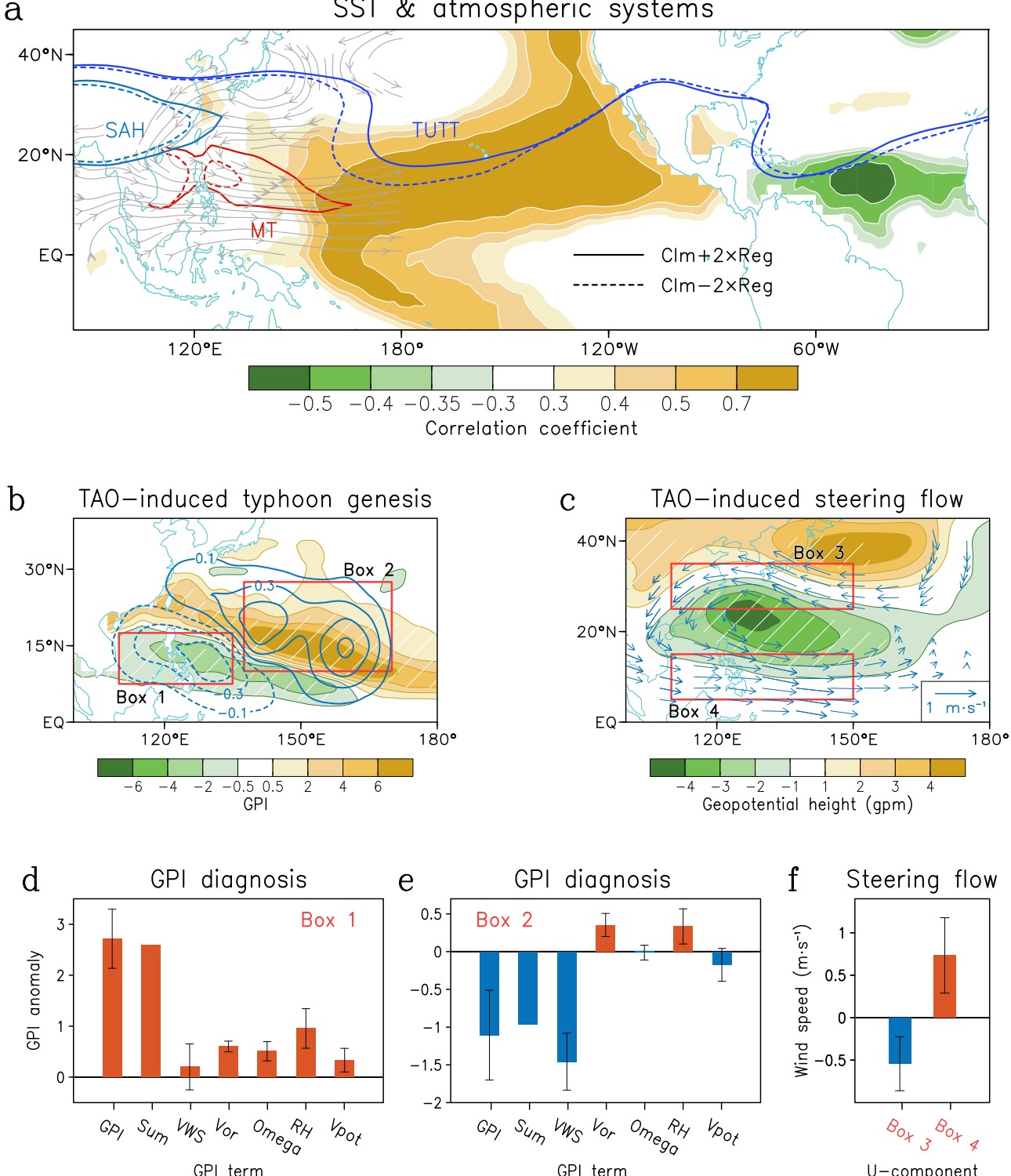

**Fig. 3 | Tropical-leaning Atlantic Oscillation (TAO) modulations on typhoon genesis and track. a** Shading in the Pacific (Atlantic) is the correlation pattern of July–September surface seawater temperature (SST) onto the simultaneous Pacific Meridional Mode (PMM) index (April–June North Atlantic Tripole, NAT, index) during 1979–2023. Streamlines are the regressed 850-hPa horizontal wind onto the PMM + NAT index. Three atmospheric systems are shown: monsoon trough (MT; represented by the relative vorticity of $3 \times 10^{-6}$ s$^{-1}$ at 850-hPa), tropical upper-tropospheric trough (TUTT; represented by the 12,420-gpm geopotential height at 200-hPa) and South Asian high (SAH; represented by the 12,500-gpm geopotential height at-200 hPa). Solid (dashed) contours are calculated as the climatology plus (minus) the double regressions onto the PMM + NAT index. **b** Regressions of peak season Genesis Potential Index (GPI; shading) and typhoon genesis density (contour; $10^{-1}$ yr$^{-1}$) on the spring TAO index. **c** Regressions of peak season 700-hPa geopotential height (shading) and steering flow (vector; $p < 0.05$) onto the spring TAO index. Hatching indicates that it passes the 95% confidence level. Regional mean GPI diagnosis in the **d** Box 1 and **e** Box 2 (outlined in **b**). Bars show the GPI anomalies and contributions from each term, with error bars indicating the 95% confidence intervals. "Sum" denotes the sum of VWS term, Vor term, Omega term, RH term, and Vpot term. **f** Zonal steering flow anomalies in Box 3 and Box 4 (outlined in **c**) regressed onto the spring TAO index.

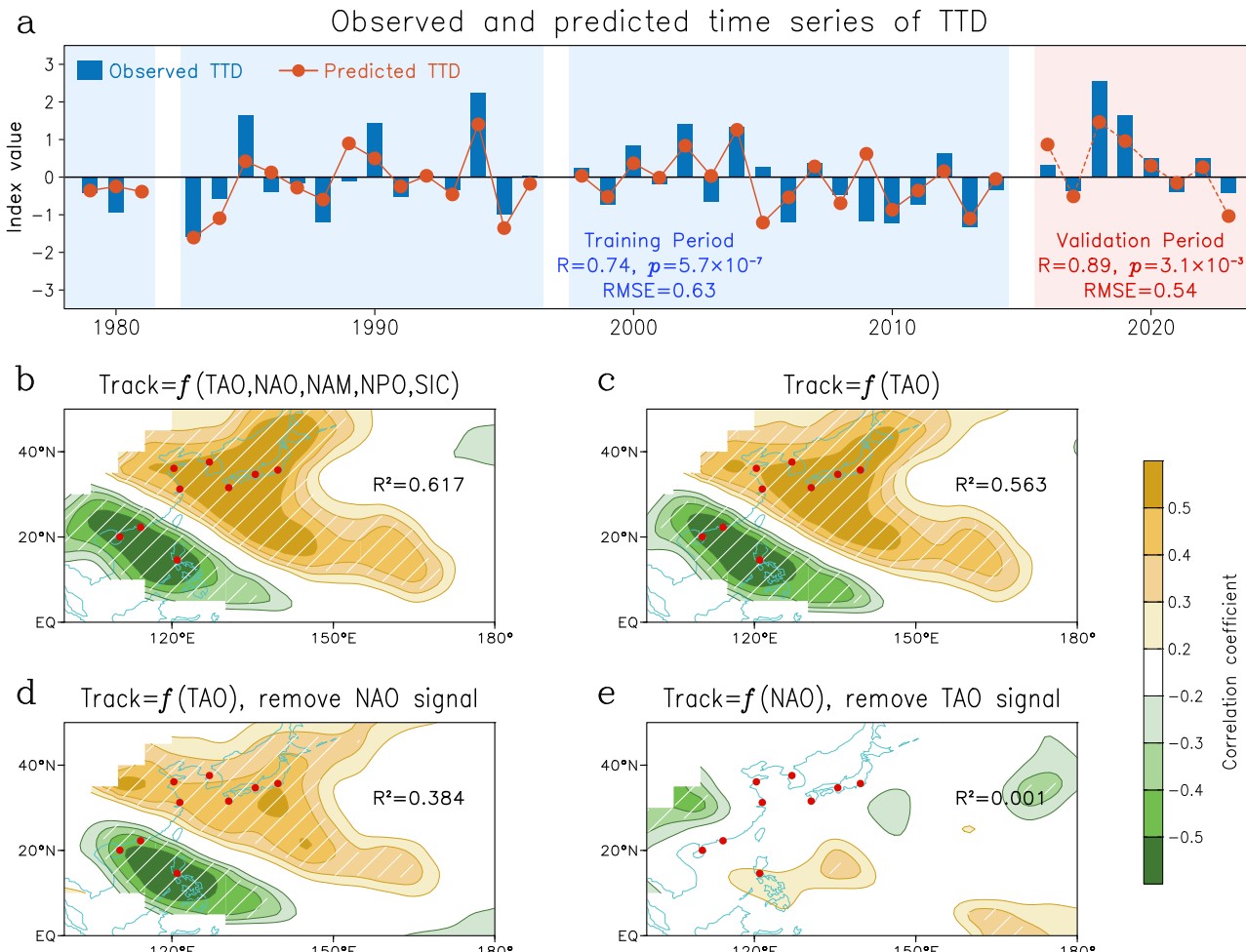

**Fig. 4 | Predicting Asia–Pacific typhoon activity. a** Time series of standardized observed typhoon track dipole (TTD) (blue bar) and predicted TTD (red line). Prediction is divided into training period (1979–2014; blue shading) and validation period (2016–2023; red shading). The correlation and significance level between observation and prediction are marked, also the root mean square error (RMSE) for two periods. **b–d** Correlation pattern between typhoon track density and time series predicted by different predictors (see "Methods" for details) in 1979–2023: using Tropical-leaning Atlantic Oscillation (TAO), North Atlantic Oscillation (NAO), Northern Annular Mode (NAM), North Pacific Oscillation (NPO) and sea-ice concentration (SIC) (**b**), TAO only (**c**), TAO only and excluding NAO signal (**d**), NAO only and excluding TAO signal (**e**). Hatching indicates that the correlation passes the 95% confidence level. The corresponding $R^2$ between TTD index and prediction model is marked.

model (Supplementary Fig. 19) reflects very low predictability ($R^2 = 14.7\%$). Moreover, when the TAO signal is removed, the predicted variances from the extratropical NAO decrease to approximately 0% (Fig. 4e), suggesting that the relationship between the extratropical NAO and TTD is not independent of the TAO. In other words, the majority of TTD predictability stems from the TAO. Hence, the TAO-model most closely reflects the Prediction$_{TTD}$ (Fig. 4c versus b, d). Above results highlight the dominant role of TAO in shaping the poleward migration of typhoon activity, enabling us to prepare for typhoon disasters up to about 4 months in advance.

In addition to the above climate variables, SST variability often has a significant influence on typhoons. However, unlike metrics such as typhoon genesis location, frequency or intensity[45,49–53], the relationship between other SST variability and TTD remains unexplored. Therefore, we further calculate the correlation coefficients between the winter-spring SST indices and the TAO as well as the TTD. As shown in Supplementary Table 2, there was no significant linear correlation between TAO (or TTD) and other existing indices (Niño3.4, PDO, AMO, and Indian Ocean Dipole). This suggests that the influence of the TAO on the TTD is generally independent of these SST modes. These findings also highlight the unique predictive significance of the TAO for the northward shift of East Asian typhoons.

Above results demonstrate that the TAO provides physically grounded and statistically independent information for predicting typhoon track variability, enriching and deepening our understanding in East Asian typhoon activity besides other major climate modes. Future work may focus on TAO influence in operational dynamical forecast models to compare the ability of TAO in forecasting typhoon track patterns.

**Projected increase of TAO events**

Beyond its role in influencing the current typhoon activity, the TAO also offers insights into future typhoon climate changes. Using the Coupled Model Inter-comparison Projection Phase 6 (CMIP6)[54] multi-member ensemble (CMIP6-MME) with historical and Shared Socioeconomic Pathway 5-8.5 (SSP585)[55] scenario experiments, we further explore the future changes in TAO. As shown in Fig. 5a, the CMIP6 models generally (23 out of 26 models) project an increase (~39.4%) in positive TAO events during the 21st century (23.96 ± 3.28 events per 100 years), compared to that during the 20th century (17.19 ± 1.85 events per 100 years). Moreover, the TAO mode obtained from the CMIP6-MME well mirror the observed one (Figs. 5b versus 1b; with a robust pattern correlation, $R = 0.92$).

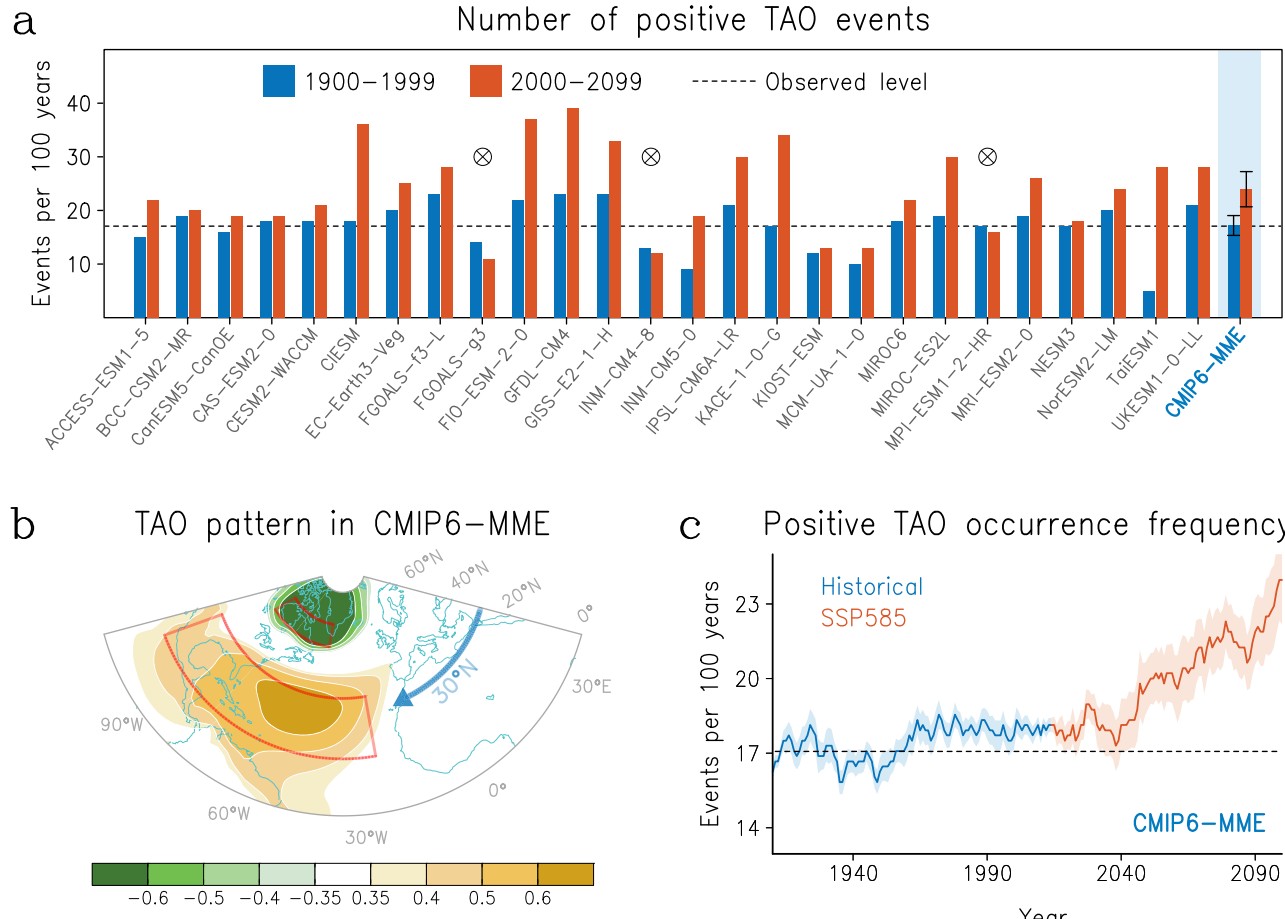

**Fig. 5 | Tropical-leaning Atlantic Oscillation (TAO) in CMIP6 multi-member ensemble. a** Comparison of positive TAO numbers (events per 100 years) over 1900–1999 (blue bars) and 2000–2099 (red bars) in Coupled Model Inter-comparison Projection Phase 6 (CMIP6) historical and Shared Socioeconomic Pathway 5-8.5 (SSP585) models. Multi-member ensemble (MME) is in the shaded background with error bar showing the 95% confidence interval. The horizontal dashed line indicates the currently observed level. Cross-in-circle marks the models that do not simulate an increase in positive TAO number. **b** Spatial correlation pattern of TAO in CMIP6 sea level pressure (SLP) during 1900–2099. **c** Evolution of positive TAO occurrence (per 100 years) diagnosed in 60-year sliding window in CMIP6 historical (blue) and SSP585 (red) ensembles. Shading is the 95% confidence interval.

The increase in positive TAO event is most pronounced toward the end of the 21st century, exceeding 20 events per 100 years (Fig. 5c). Moreover, there are 7 out of 26 CMIP6 models can even capture the strengthening cross-seasonal linkage between the WNP anomalous cyclone and preceding spring TAO (Supplementary Fig. 20), further implying more prominent impacts on typhoon track potentially from TAO as the climate warms. In addition, the future anthropogenic warming would decelerate typhoon motions near high-latitude populated coastal regions due to a poleward shift of the subtropical westerlies, which can induce a longer time of typhoon impacts[56]. And accompanying with increasing intensity and rainfall[57], this can potentially compound future typhoon-related damages. Based on the current results, the projected increase in positive events of TAO may potentially contribute to increases in steering-flow patterns conducive to poleward migration of typhoon tracks. Such circulation changes imply the possibility that high-latitude cities in East Asia would face more typhoon threats and relevant economic losses in the future. Therefore, here we boldly infer that the poleward migration of typhoon activity induced by TAO in a warming future would exacerbate the societal risk from typhoons in East Asian high-latitude cities as the century progresses.

It should be noted that our inference is based on the premise that the linkage between the TAO and TTD remains unchanged. Since the cross-seasonal linkage between the two keeps unchanged even after excluding the long-term trends (Supplementary Figs. 15 and 16). However, under the context of global warming, typhoon activity and tropical SST patterns are evolving, thus uncertainty is always an inevitable existence. For instance, the frequency of future typhoons may continue to decrease with global warming[58–61], which might alter the response of typhoon activity to TAO indirectly. Besides, CMIP6 outputs used here do not simulate typhoon tracks directly, and steering-flow changes alone cannot fully translate to future track density changes because the latter are also strongly affected by shifts in TC genesis location. Furthermore, considering that the influence of TAO on WNP circulations depends on teleconnection processes between the ocean and atmosphere, the response of WNP typhoon activity to the projected changes in TAO might be sensitive to the global warming-induced SST patterns in the tropics[62], and thereby leading to uncertainty in the cross-seasonal TAO-TTD linkage.

We similarly explored future changes in the frequency of positive phase NAO events (Supplementary Fig. 21). However, the multi-model results show a significant decrease in the frequency of positive NAO events. Considering the positive correlation between NAO and TTD (Fig. 2a), this contradicts the poleward shift of typhoon activity as climate warms. Therefore, the relationship between NAO and typhoon in the current observations may not be physically reliable, but rather mediated through TAO. However, the physical relationship between

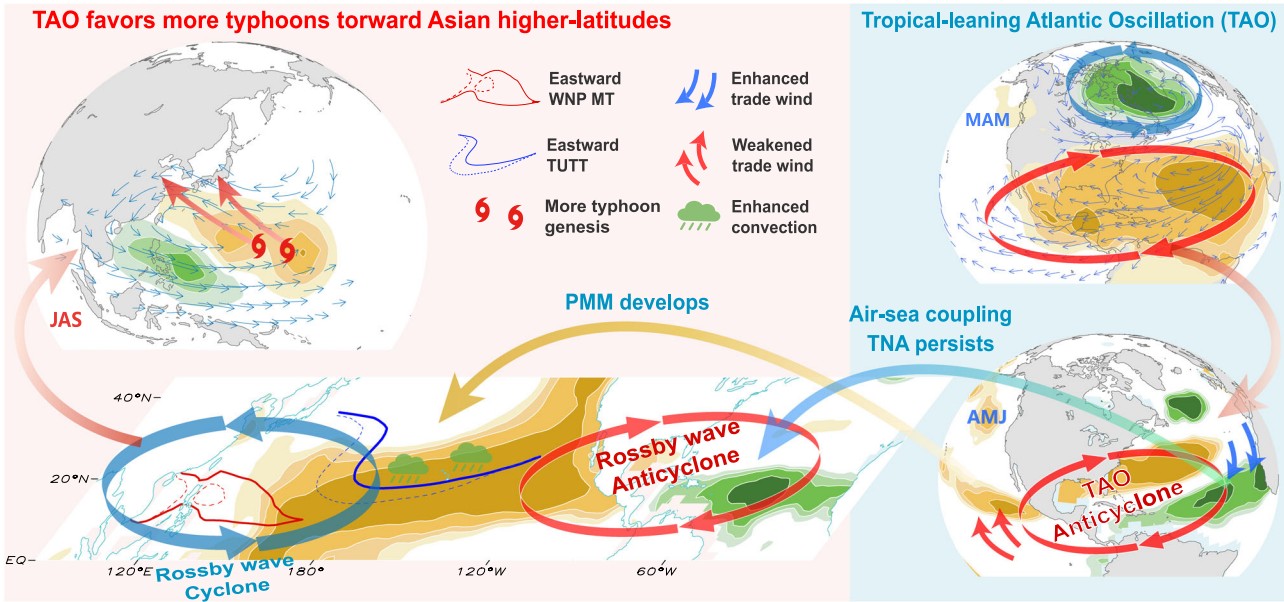

**Fig. 6 | Schematic diagram illustrating the influence of the Tropical-leaning Atlantic Oscillation (TAO) on Northwest Pacific typhoons.** During spring (March–May; MAM), the TAO weakens trade winds over the subtropical north-eastern Pacific, leading to local surface seawater temperature (SST) warming and subsequent (e.g., April–June; AMJ) development of the Pacific Meridional Mode (PMM) through the seasonal footprint mechanism. Concurrently, TAO strengthens the trade winds over the tropical North Atlantic (TNA), resulting in SST cooling that persists into summer (July–September; JAS) via local air-sea coupling. The cold TNA SST anomalies can continuously induce Rossby wave anticyclone to its west, thereby further promoting PMM development. The Atlantic-eastern Pacific Rossby wave anticyclone has been demonstrated to directly favor the Northwest Pacific cyclone via Pacific convection, a phenomenon that is explained by the Pacific–Atlantic subtropical oscillation (PASO)[42]. Consequently, the presence of a lower-level cyclone and an upper-level anticyclone in the Northwest Pacific leads to an eastward shift in the tropical upper-tropospheric trough (TUTT) and monsoon trough (MT) (solid line for positive TAO and dashed line for negative TAO), promoting more (less) frequent typhoon genesis in the eastern (western) Northwest Pacific. The altered steering flow favors (inhibits) typhoon movement towards higher (lower) latitudes of East Asia.

TAO and classical extratropical NAO requires more detailed investigations.

While we suggest that CMIP6 models show a robust projection of more frequent positive TAO events, some uncertainties remain. The spread among models primarily arises from differences in simulating the mean-state and variability of North Atlantic SSTs, as well as the subtropical jet response[63,64]. It is reported that most CMIP6 models exhibit cold SST biases in the North Atlantic, especially in the tropics[65,66]. This may cause an overestimation of positive TAO frequency because positive TAO corresponds to negative TNA SST anomaly. Nevertheless, the overall positive shift in TAO occurrence is consistent across most models, indicating that the projected intensification of positive TAO is a robust multi-model signal rather than an artifact of a few outliers. Future work could further assess the sensitivity of TAO projections to Atlantic SST biases and internal variability through targeted model experiments.

## Discussion

Existing studies have highlighted the significant influence of extratropical NAO on tropical climate[67]. For instance, the extratropical NAO has been linked to the NAT SST pattern through local air-sea coupling processes[68,69], which can induce persistent SST anomalies in the TNA and subsequently affect the development of ENSO events[70]. On decadal timescales, signals of the extratropical NAO are found to drive changes in the Atlantic meridional overturning circulation through the sea surface wind stress[71] and precede the Atlantic Multidecadal Oscillation by 15–20 years[72]. Additionally, NAO can modulate tropical climate backgrounds and contribute to unstable TNA-ENSO relationships[33]. In contrast, the TAO identified in present study, being situated closer to the tropics (Fig. 1a, b), naturally exerts a more direct influence on tropical climate anomalies, as shown by our comparative

analysis that TAO possess much stronger effects on WNP typhoon activity compared to the classical extratropical NAO (Figs. 2a and 4). Therefore, TAO offers a promising and unique predictor for tropical climate variability. A schematic diagram is presented in Fig. 6.

The nature of climatic oscillations, and the changes in their behaviors, including predictability of SST, steering flow, or convective rainfall patterns with a few months lead time, are central to understanding peak season typhoon poleward migration. The spring TAO mode provides valuable early warning for humans to plan and take adaptive measures to minimize loss of life and property during the active summer typhoon season in East Asian coastal cities. This study emphasizes one of several often-unheeded aspects of typhoon variability in response to climatic atmospheric variability. Besides, given that the PMM plays a critical role in shaping ENSO diversity[73,74], the prominent physical linkages between TAO and PMM further underscore the importance of TAO. Hence, the proposal of TAO not only enhances the predictability of seasonal typhoon activity but also opens a window for future research on exploring tropical climate prediction, including its potential to improve the forecast skill of pan-tropical interactions and ENSO-related variability. Nonetheless, this work has only initially explored the physical mechanisms and climatic effects of TAO. Therefore, more in-depth studies are needed in the future.

## Methods

### Typhoon data and processing

We focus on the East Asia–WNP typhoon (TC with $V_{max} > 32.6\,m\,s^{-1}$) during July–September of 1979–2023, using the Joint Typhoon Warning Center TC best-track datasets. Although TAO exerts some influence on typhoon activity in other months, its relationship with the TTD mode is mainly concentrated in the peak season (i.e., July–September) when WNP typhoon are most active (Supplementary Fig. 22). Thus, we mainly

focus on July–September. This study primarily focuses on typhoons because stronger typhoons are more likely to cause significant damage[59,75]. We also examined results for all TCs with $V_{max} > 17.2 \, \text{m s}^{-1}$, which showed consistent findings (Supplementary Fig. 23), although the relationship with TAO may be somewhat weaker for weaker TCs due to their inherent randomness. The genesis (marked by $V_{max} > 17.1 \, \text{m s}^{-1}$) density and track density are calculated by interpolating typhoon passage onto 5° longitude × 5° latitude grid points. A 9-point spatial smoothing is further applied to the density data because the influence of typhoon exists over a wide range[76]. The ACE is the square of the maximum wind speed[31]. The EOF method was applied to the normalized track density in (100°–145°E, 10°–45°N) to identify the TTD mode. We removed the signal of the original first EOF mode from all data via linear regression because it is mainly a climatic signal, with little anomaly to be predicted (Supplementary Figs. 24 and 25). We also use the TC best-track data from the Japan Meteorological Agency and the Shanghai Typhoon Institute of the China Meteorological Administration for validation.

### Reanalysis data

We primarily use the monthly atmospheric reanalysis version 5 data from the European Center for Medium-range Weather Forecasts (ERA5, resolution of 2.5° × 2.5°)[77]. The SST data (resolution of 2° × 2°) are merged from the NOAA Extended Reconstructed SST version 5 (ERSSTv5)[78] and the Hadley Centre Global SST version 1.0 (HadISST1)[79]. Three extreme El Niño years (1982, 1997, and 2015) are excluded, which may disturb the TAO-TTD relationship. However, we suggest that ENSO is likely to play a limited role in this relationship (Supplementary Fig. 6 and Supplementary Table 2). Including ENSO years will not significantly alter our conclusions (Supplementary Table 1). Steering flow is the pressure-weighted horizontal wind averaged from 850-hPa to 300 hPa. The significance of Pearson correlation and linear regression is evaluated by the two-tailed Student's $t$ test.

### Climate indices

This paper employs multiple climate indices, primarily atmospheric circulation mode, SIC and SST indices, with specific definitions provided in Supplementary Table 3. We define the spring (March–May) TAO and NAO indices based on the difference in SLP between two regions. We further compare our NAO index with the downloaded NAO index (see "Data availability"), which is the leading EOF mode of SLP in (90°W–40°E, 20°N–80°N), and find that the two time series show strong Pearson correlation ($R = 0.97$, $p < 0.001$) during spring. The SST index is defined as the regional average surface seawater temperature. We utilized multiple precursors from preceding winter and spring seasons (with specific seasons determined by high correlation coefficients in Supplementary Fig. 26) to highlight the unique and dominant role of TAO.

### Genesis potential index

The GPI[46] is used to diagnose the modulation effects of environmental conditions on the typhoon genesis:

$$GPI = \left| 10^5 \eta \right|^{1.5} \left( \frac{RH}{50} \right)^3 \left( \frac{V_{pot}}{70} \right)^3 \left( 1 + 0.1 V_s \right)^{-2} \left( 1 - 10\omega_{500} \right) \quad (1)$$

where $\eta$ is the 850-hPa absolute vorticity, RH the 700-hPa relative humidity, $V_{pot}$ the maximum potential intensity, $V_s$ the vertical wind shear between 200 hPa and 850 hPa, and $\omega_{500}$ the 500-hPa vertical velocity.

$V_{pot}$ is calculated as:

$$V_{pot} = \sqrt{ C_p \frac{C_K}{C_D} \frac{T_s}{T_0} \left( T_s - T_0 \right) \left( ln\theta_e^* - ln\theta_e \right) } \quad (2)$$

Where $C_p$ is the specific heat capacity at constant pressure, $C_K$ the air-sea heat circulation exchange coefficient, and $C_D$ the sea surface drag coefficient. $T_s$ sea surface temperature, $T_0$ the average temperature of the outflow layer, $\theta_e^*$ the sea surface saturation equivalent potential temperature, and $\theta_e$ the boundary layer equivalent potential temperature.

$V_s$ is calculated as:

$$V_s = \sqrt{ \left( u_{200} - u_{850} \right)^2 + \left( v_{200} - v_{850} \right)^2 } \quad (3)$$

Where $u$ is the zonal wind, $v$ is the meridional wind, and the subscript indicates the pressure level (hPa).

To attribute GPI anomalies associated with the TAO to individual environmental factors, a linear decomposition was applied as follows: for each environmental factor, the corresponding TAO-related anomaly was substituted into the GPI calculation, while all other variables were retained at their climatological states. The total GPI anomaly was subsequently approximated as the sum of the contributions from all factors. This approach enables the relative roles of vertical wind shear (VWS), low-level relative vorticity (Vor), mid-level vertical velocity (Omega), mid-level relative humidity (RH), and potential intensity (Vpot) in modulating TC genesis potential to be quantitatively assessed.

### Wave activity flux

We use the wave activity flux (WAF)[80] to effectively analyze the propagation characteristics of anomalously steady Rossby waves. The horizontal component of WAF is calculated as:

$$WAF = \frac{p}{2000|\mathbf{U}|} \begin{pmatrix} U\left(\psi'^2_x - \psi'\psi'_{xx}\right) + V\left(\psi'_x\psi'_y - \psi'\psi'_{xy}\right) \\ U\left(\psi'_x\psi'_y - \psi'\psi'_{xy}\right) + V\left(\psi'^2_y - \psi'\psi'_{yy}\right) \\ \frac{f_0^2}{N^2}\left[ U(\psi'_x\psi'_z - \psi'\psi'_{xz}) + V(\psi'_y\psi'_z - \psi'\psi'_{yz}) \right] \end{pmatrix} \quad (4)$$

Where $p$ is the pressure, $\psi'$ is the perturbation streamfunction relative to climatology, $f_0$ is the Coriolis parameter, $N$ is the static stability, and $U$ and $V$ are the climatology of zonal and meridional winds, respectively. $|\mathbf{U}|$ is the magnitude of $(\vec{U,V})$. The subscripts $x$ and $y$ represent the zonal and meridional gradients, respectively.

### Multiple linear regression model

Multiple linear regression model can be expressed as:

$$M_{reconstruction} = \beta_0 + \sum_{i=1}^{n} \beta_i N_i \quad (5)$$

where $\beta_0$ is the intercept, $\beta_i$ are regression coefficients, $M_{reconstruction}$ is the reconstructed index, $N$ is the index of variables used. The coefficients, in the form of matrix $\boldsymbol{\beta}$, are estimated using the ordinary least squares method, minimizing the residual sum of squares:

$$\boldsymbol{\beta} = \left( \mathbf{N}^T \mathbf{N} \right)^{-1} \mathbf{N}^T \mathbf{M} \quad (6)$$

$$\beta_0 = \bar{M} - \sum_{i=1}^{n} \beta_i \bar{N}_i \quad (7)$$

where $\mathbf{N}$ is the design matrix including all variables, and $\mathbf{M}$ is the response vector. The upper horizontal line indicates the mean value.

### CMIP6 data and processing

The CMIP6 historical and SSP585 simulations are used[54,55]. We use 26 models (Supplementary Table 4) with historical experiments for 1850–2014 and SSP585 experiments for 2015–2100. For each model,

**Article** https://doi.org/10.1038/s41467-025-67946-4

only the first ensemble member available is selected. Most are r1i1p1f1, except for MIROC-ES2L and UKESM1-0-LL (r1i1p1f2). Data from different model resolutions are interpolated into the same horizontal resolution of 2.5° × 2.5°. We use a sliding climatology of 60 years to standardize the TAO index. For the first 60 years (1850–1909), climatology was fixed at 1850–1909. For years after 1910, the climatology was defined as the 60 years before that year. We use a threshold of +1.0 standard deviation to detect positive TAO events in CMIP6 model outputs. The definition of the TAO and NAO indices in models are the same as in observations.

## Data availability
The JTWC TC best-track data are available at https://www.metoc.navy.mil/jtwc/jtwc.html?best-tracks, the CMA TC best-track data are available at https://tcdata.typhoon.org.cn/en/, the JMA TC best-track data are available at https://www.jma.go.jp/jma/jma-eng/jma-center/rsmc-hp-pub-eg/besttrack.html, the ERA5 data are available at https://www.ecmwf.int/en/forecasts/dataset/ecmwf-reanalysis-v5, the NOAA ERSSTv5 dataset are available at https://www.esrl.noaa.gov/psd/data/gridded/data.noaa.ersst.v5.html, the HadISST1 data are available at https://www.metoffice.gov.uk/hadobs/hadisst, the simulation output data of CMIP6 are available at https://esgf-node.llnl.gov/search/cmip6, the downloaded NAO index is available at https://climatedataguide.ucar.edu/climate-data/hurrell-north-atlantic-oscillation-NAO-index-pc-based.

## Code availability
Figures were plotted with the Grid Analysis and Display System v2.2.1, available at http://opengrads.org/. The source codes used to produce these main results are available from https://doi.org/10.5281/zenodo.13901572.

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

## Acknowledgements

We thank the State Key Laboratory of Severe Weather Meteorological Science and Technology (LaSW) and Professor Peiliang Li at Zhejiang University for providing support and help. This work was jointly supported by grants from: the Guangdong Major Project of Basic and Applied Basic Research (2020B0301030004), the National Natural Science Foundation of China (42405034), the Southern Marine Science and Engineering Guangdong Laboratory (Zhuhai) (SML2024SP012), the Innovation Group Project of Southern Marine Science and Engineering Guangdong Laboratory (Zhuhai) (No. 311024001), the Guangdong Province Key Laboratory for Climate Change and Natural Disaster Studies (2023B1212060019), the Shanghai Typhoon Research Foundation (TFJJ202506), and the Zhejiang University "Ocean College Seed Fund": Excellent Young Teachers Training Project (2025YQ001). We are grateful for the research start-up funding support of the "Top 100 Talents Plan" Project of Zhejiang University and the high-performance computing conditions of the Ocean College at Zhejiang University.

## Author contributions

Conceptualization and supervision: C.H. Investigation, visualization and writing-original draft: Z.W., C.H. Methodology: Z.W., C.H., L.L. Writing-review and editing: Z.W., C.H., L.L., W.C., T.L., G.H., C.Z., R.W., and D.C.

## Competing interests

The authors declare no competing interests.

## Additional information

¹State Key Laboratory of Ocean Sensing & Ocean College, Zhejiang University, Zhoushan, China. ²Shanghai Typhoon Institute, China Meteorological Administration, Shanghai, China. ³Southern Marine Science and Engineering Guangdong Laboratory (Zhuhai), Zhuhai, China. ⁴Laoshan Laboratory, Qingdao, China. ⁵Key Laboratory of Physical Oceanography/Frontier Science Centre for Deep Ocean Multispheres and Earth system, Ocean University of China, Qingdao, China. ⁶State Key Laboratory of Marine Environmental Science & College of Ocean and Earth Sciences, Xiamen University, Xiamen, China. ⁷State Key Laboratory of Loess and Quaternary Geology, Institute of Earth Environment, Chinese Academy of Sciences, Xi'an, China. ⁸Climate Center, Guangxi Meteorological Bureau, Nanning, China. ⁹State Key Laboratory of Satellite Ocean Environment Dynamics, Second Institute of Oceanography, Ministry of Natural Resources, Hangzhou, China. ¹⁰Key Laboratory of Earth System Numerical Modeling and Application, Institute of Atmospheric Physics, Chinese Academy of Sciences, Beijing, China. ¹¹College of Earth and Planetary Sciences, University of Chinese Academy of Sciences, Beijing, China. ¹²School of Earth Sciences, Zhejiang University, Hangzhou, China. ✉e-mail: hucd@zju.edu.cn

