## [Transparent Peer Review File · Nature Communications]

Tropical-leaning Atlantic Oscillation favors more typhoons toward Asian high-latitude cities

Corresponding Author: Dr Chundi Hu

Version 0:

Reviewer comments:

Reviewer #1

(Remarks to the Author)

Comments on “A new climate oscillation takes typhoon activity toward Asian high-latitude cities” by Wu et al. This study focuses an important issue of atmospheric mode on NWP typhoons. In general, this manuscript is well written and shows a clear story line of how TNAO impacts NWP typhoons. However, I really concern the novelty and robustness, which could need extra analyses.

Major Comments:

1. The authors show the TNAO impact on tropical cyclone over the NWP. However, I find that the pathway TNAO impact is much similar to the North Tropical Atlantic (NTA) SST impact on Pacific climate (Ham et al. 2013). In this sense, I considered that the TNAO and NTA SST are air-sea coupled system that they share the same physical process affecting the NWP Typhoon. Moreover, the TNAO is relatively transient compared with the NTA and the statistical covariance is fragile. Therefore, the verification of TNAO impact on NWP typhoon activity in numerical models is of great importance.
2. The authors considered the poleward migration of NWP typhoon under global warming. However, many studies have shown that the NWP typhoon frequency will decrease under the global warming (Knutson et al. 2020). The CMIP6 models can only reflect the TNAO changes, however, the connections between NWP typhoons and TNAO may also change. The changed NWP typhoons may also shape the NWP typhoon – TNAO relationship.
3. The authors only considered the typhoons, which reach 32.6m/s and above, right? However, I believe that the TNAO could affect both typhoons and tropical cyclones (17-32.6 m/s) in the NWP. The authors need to show the TNAO impact on all TC frequency and explain why they only focused on typhoons?

Minor comments:

1. Line 299: The definition of jet shift index could be incorrect. Please modify the definition region. Also, it is worthy noted that the authors defined too many atmospheric modes in the Method part, I suggested to revise the definitions and explain the purpose of the usage.
2. It is unfair to move the TNAO signal from the NAO index since they are highly correlated in Fig.4 . The authors could also check the TNAO-TC track correlation when removing the NAO signal. I believe it is also weak and insignificant.
3. Since the TNAO is induced by the jet shear, I also want to the correlations between North America jet stream and NWP typhoons (and TNAO).
4. Line 126: why do they authors only remove three strong El Niño events? I believe that the TNAO is highly correlated with the NTA SST simultaneously and consequently ENSO signal. Therefore, it may also need to consider both El Niño and La Niña events.
5. Line 273 and Line 73: the typhoon season is defined as the months from July to September or August? Note, the portion of July-September typhoons could not be a majority for the whole typhoon season tropical cyclone genesis frequency from June to October.

References:

Ham et al. Sea surface temperature in the north tropical Atlantic as a trigger for El Niño/Southern Oscillation events
Knutson et al. Tropical Cyclones and Climate Change Assessment: Part II: Projected Response to Anthropogenic Warming

(Remarks on code availability)

Reviewer #2

(Remarks to the Author)

Review of Manuscript "A New Climate Oscillation Takes Typhoon Activity Toward Asian High-Latitude Cities"

This manuscript presents a novel climate oscillation, the tropical-leaning North Atlantic Oscillation (TNAO), and explores its predictive capability for the poleward migration of typhoon activity in East Asian high-latitude cities. The study makes a significant contribution to the field by identifying a distinct oscillation from the classical North Atlantic Oscillation (NAO) and demonstrating its robust predictive skill for typhoon tracks. The findings are particularly relevant for improving seasonal forecasts and disaster preparedness in the region under the background of global climate change.

The identification of the TNAO as a distinct oscillation from the NAO is a major contribution. The results and conclusions in study are basically convincing. Also, the use of multiple datasets (e.g., ERA5, CMIP6) and statistical techniques (e.g., EOF analysis, wave activity flux) strengthens the credibility of the findings. The exclusion of extreme El Niño years enhances the robustness of the TNAO-TTD relationship. This study clearly links TNAO to jet stream variability, SST anomalies (PMM, NAT), and subsequent typhoon steering flows. The cross-basin teleconnection mechanism is well-articulated.

Major comments:

(1) While the distinction from NAO is emphasized, further discussion is needed on why TNAO was not identified earlier. A spectral analysis or comparison with other known modes (e.g., Atlantic Multidecadal Oscillation) could solidify its uniqueness.

(2) Please clarify whether TNAO is independent of ENSO on longer timescales, as only extreme El Niño years were excluded.

(3) The Gill-type Rossby wave response and "subtropical Atlantic-Pacific teleconnection" require more detailed explanation or references. A schematic diagram could aid understanding.

(4) Please address how TNAO-induced SST anomalies persist into summer. Is this due to ocean memory or atmospheric feedback?

(5) About the CMIP6 Projections: the 39.4% increase in positive TNAO events is intriguing but lacks discussion on model spread or uncertainties. How do model biases (e.g., in Atlantic SST) affect this projection?

Other comments and suggestions:

(1) The statement on decelerated typhoon motion seems tangential; either expand or omit.

(2) The abstract overstates "inevitably face more serious threats." Modify to reflect uncertainties in CMIP6 projections.

(3) Discuss how TNAO's predictive skill compares to existing indices (e.g., PDO, AMO).

(4) Highlight limitations, such as the assumption of stationary TNAO-typhoon relationships under climate change.

Generally, this study makes a compelling case for the TNAO as a key driver of typhoon variability. With minor revisions to clarify mechanisms and contextualize findings, it would be a valuable contribution to high-impact journals. The reviewer recommends acceptance after addressing the above points.

(Remarks on code availability)

Reviewer #3

(Remarks to the Author)

Summary: The manuscript identifies and characterizes a novel springtime North Atlantic seesaw—termed the Tropical-leaning North Atlantic Oscillation (TNAO)—and demonstrates its predictive linkage to summer typhoon-track density (TTD) over the western North Pacific via modulation of the Pacific Meridional Mode (PMM). By defining a spring sea level pressure (SLP) difference index, the authors show robust correlations ($R \approx 0.75$) with the leading empirical orthogonal function (EOF) of typhoon tracks, far exceeding the classical NAO's skill. They propose two cross-basin pathways: (1) TNAO-driven anticyclonic flow over the Northeastern Pacific weakens trades, warms SST, and seeds the PMM; and (2) an Atlantic SST tripole forces a Rossby response that reinforces the PMM. CMIP6 projections under SSP5-8.5 indicate increases in positive TNAO events, suggesting enhanced future typhoon poleward migration. Overall, the conceptual advance is substantial, linking two ocean basins through a physically plausible teleconnection with seasonal forecasting potential.

General evaluation: This study presents a new cross-basin teleconnection with strong predictive implications, making a substantial and well-validated contribution to our understanding of cross-basin climate-typhoon linkages and offering a seasonal forecasting potential. Pending clarifications on several issues described below and some extended analysis to strengthen the robustness, I recommend acceptance after a major revision. I believe that once these points are satisfactorily resolved, the paper will be well suited for Nature Communications.

Major Comments

1. Asian high-latitude cities: In the title, the authors emphasized the typhoon activity toward Asian high-latitude cities. However, throughout the manuscript, the reviewer could not find any analysis on landfalling typhoons that threatened cities in the relevant regions. The authors should explicitly demonstrate the point by showing the increase in landfall frequency. An analysis of landfall frequency or distance from coast would help clarify the claim.
2. Long-term trend vs internal climate variability: Given global warming, both Atlantic and Pacific climates have a warming trend. Have the authors removed a linear (or higher-order) trend from their SLP/SST fields before computing TNAO and PMM? Note that a lingering trend could spuriously enhance their teleconnection statistics.
3. Memory from prior seasons: Spring TNAO events may be influenced by preceding winter's modes (e.g., ENSO, IOD, PDO, AMO). Ensuring the predictor is not contaminated by autoregressive persistence will clarify the true lead-time benefits.
4. Pacific-basin wind-anomaly: The manuscript shows only the Atlantic (NAT/TNAO) 850 hPa anomaly map (Supplementary

Fig. 5a), but omits the analogous Pacific anomaly field demonstrating trade-wind weakening. Please include a spatial map of 850 hPa wind anomalies regressed onto the PMM (or TNAO) over the Pacific to visually confirm the westerly anomalies in the 8–18° N band.

5. Confounding climate modes: Other low-frequency modes (AMO, PDO/IPO), ENSO, and IOD may influence both TNAO and typhoon metrics, potentially biasing the TNAO→TTD link. The authors should perform multivariate or partial-correlation tests that control for AMO, PDO/IPO, ENSO (Niño 3.4), and IOD indices and to confirm that TNAO retains unique predictive skill.

6. Potential confounding from monsoon and shear: Summer monsoon trough position and vertical wind shear directly modulate both genesis and track of typhoons over the western North Pacific. Even if PMM explains part of the signal, the residual may simply be a shifted monsoon trough or the Pacific TUTTs. Could the authors provide any insights on possible connection? At least, this should be discussed briefly to demonstrate the PMM's unique pathway as discussed in the manuscript.

7. Regression-training sample size and robustness: Training the TNAO→TTD regression on 1979–2014 (36 yr) risks sampling uncertainty; the validation period (2016–2023) showing higher R^2 may reflect lucky sampling or non-stationarity. The authors can conduct sensitivity tests (e.g., leave-one-decade-out, bootstrap/jackknife) to quantify confidence intervals and show windowed or running regressions over 1979–2023 to assess temporal stability of the linkage.

8. Mechanistic quantification: The proposed pathways (thermodynamic PMM vs. direct circulation steering) are described qualitatively. Could the authors provide a quantitative analysis to show the relative importance of thermodynamic and dynamic factors in the GPI analysis and quantify the consistent steering change? The authors may also include precipitation/OLR composites to verify convective amplification under the PMM.

9. Operational predictability: To underscore novelty, showing how a simple TNAO-based statistical forecast compares against state-of-the-art dynamical seasonal typhoon outlooks (e.g., ECMWF, GFS) in hindcast mode would demonstrate added value of the finding. Does the regression model outperforms the existing operational models?

10. Physical process diagrams or schematic: Finally, a concise schematic that lays out both the direct anticyclone-driven trade weakening and the Atlantic-tripole/Gill-mode response pathway would help readers keep track of the multiple legs of the cross-basin teleconnection.

11. Translation from projected TNAO to northward shift of typhoon activity: The authors show an increased frequency of positive TNAO events under SSP5-8.5, but how that translates into societal risk (e.g. changes in ACE, landfall probability, economic exposure) remains unexplored. A brief discussion would help readers grasp the real-world implications. It will be much convincing if the authors can demonstrate the connection by analyzing typhoon activity in CMIP6 models ensemble simulations.

Minor Comments

1. The global analyses (EOFs, regressions, composites) lack a clear statement of their full temporal coverage (e.g., 1979–2023) beyond the training/validation split. Therefore, please explicitly state the data period in the Methods and all figure captions (e.g., “analyses based on 1979–2023”).

2. “high summer” vs. “peak season”: The manuscript alternates between “high summer” (July–August) and “peak season” (July–September), which can confuse readers. Please keep “peak season” since all analyses seemed to have focused on peak season.

3. Please give the rationale for excluding extreme El Niño years from the analysis and briefly discuss and explicitly show its potential impact.

4. L146–151: Clarify that the anticyclonic flow over the northeast Pacific produces westerly anomalies on its equatorward flank, vectorially opposing the northeasterly trades since this could not be seen from Supplementary Fig. 5a.

5. Lines 50-51 “While the impacts of shifting internal natural climatic oscillations on typhoon activity are often overlooked” is not true. There have been many studies that investigated the impacts of various internal long-term climate modes on typhoon activity, including PDO, IPO, AMO, etc. The authors should mention those here.

6. Although references are comprehensive, some recent work on cross-basin teleconnections where relevant should be included.

7. Line 58, remove “attribution of”.

8. Line 537, change “exerts” to “exert”.

(Remarks on code availability)

Version 1:

Reviewer comments:

Reviewer #1

(Remarks to the Author)

I believe that the authors have addressed all my concerns. I am satisfied with current revision and suggest to accept this manuscript.

(Remarks on code availability)

Reviewer #2

(Remarks to the Author)

It seems that all my concerns have been addressed in the revised MS and responses to reviewers. The reviewer felt that the MS could be accepted for publication.

(Remarks on code availability)

These codes are relatively complete and can be easily read and reproduced.

Reviewer #3

(Remarks to the Author)

Summary and Overall Assessment

The revised manuscript presents the identification of a previously unrecognized climate mode—the tropical-leaning North Atlantic Oscillation (TNAO)—and argues that this springtime SLP dipole is a strong and physically meaningful predictor of peak-season western North Pacific (WNP) typhoon track variability, particularly the poleward migration toward high-latitude East Asian cities. The authors propose two physical pathways linking TNAO to the Pacific Meridional Mode (PMM), tropical North Atlantic SST anomalies, and subsequent summertime steering flow anomalies. They also examine CMIP6 projections, suggesting more frequent positive TNAO events under SSP5–8.5.

This work proposes a novel and interesting cross-basin teleconnection with potential implications for seasonal typhoon forecasting. The revisions substantially improve the clarity, robustness, and physical justification of the results. The authors have responded carefully and extensively to reviewer comments, adding new analyses (trend-removed fields, ENSO controls, GPI decomposition, jet shift index revision, additional figures, sensitivity tests) and clarifying mechanisms. Overall, the revised manuscript is strong and now suitable for publication after addressing a few remaining issues that concern clarity, over-interpretation, and the need for more cautious framing in several places as summarized in my comments given below.

Remaining Issues to Address Before Acceptance

1. Over-extrapolation of CMIP6 TNAO projections to future typhoon risks: While the authors acknowledge limitations, some statements still imply direct future increases in typhoon hazards due to TNAO changes (e.g., Lines 324–329). CMIP6 outputs used here do not simulate typhoons directly, and steering flow differences alone cannot fully translate to future track density changes because the latter are also strongly affected by shifts in genesis locations. Furthermore, the response of TC activity over the WNP to the projected changes in TNAO might be also sensitive to the global warming-induced SST patterns in the Tropics (see a review by Wang et al. 2025), which has not been examined in this study and thus be mentioned in the discussion section. I thus suggest further soften any direct implication that increased positive TNAO would cause increased typhoon frequency or ACE near high-latitude cities and emphasize instead where applicable (including in the abstract) “... may potentially contribute to...” “suggests possible increases in steering-flow patterns conducive to poleward migration...”. Wang, Y., M. Satoh, R.-F. Zhan, J.-W. Zhao, and S.-P. Xie, 2025: Tropical sea surface warming patterns and tropical cyclone activity – A review, *Adv. Atmos. Sci.*, 42(10), 1996–2017, <https://doi.org/10.1007/s00376-025-5114-1>.
2. Clarify the added “city-box” frequency results: The added frequency increments (+1.0/–0.65) are useful but need more explanation, such as whether they are counts per season, or normalized by area, or if the two boxes are comparable in size, and the numbers should be area-adjusted. I suggest add a sentence explaining the units and comparability.
3. Potential circularity between PMM definition and TNAO-forced Pacific SST: The PMM region used in the lead-lag analysis partly overlaps with the area of westerly anomalies produced by TNAO. While this is physically plausible, it can also appear self-reinforcing. I would recommend add 1–2 sentences acknowledging this potential dependence and explaining why the interpretation remains robust.
4. Strength of the jet-TNAO relationship: The NAJS–TNAO correlation ($R \approx 0.85$) is very high, implying the two indices may be nearly redundant. The manuscript should clarify whether TNAO provides additional information beyond NAJS for typhoon prediction and why TNAO is more directly connected to steering-flow anomalies than NAJS alone. A brief statement would resolve this.

Minor Comments

1. Lines 29–32, “covariance above 56%” — specify this refers to TNAO explaining 56% of TTD variance.
2. Line 85, change to “...can directly capture the previously overlooked TNAO signal.”
3. “TUTT” appears as “tropical upper-tropospheric trough” in line 211 while “tropical upper-troposphere trough” in line 759. Use uniform terminology.

(Remarks on code availability)

Response to Reviewer's comments

We are grateful for the time and effort that the reviewers and the editorial team dedicated to evaluating our work. Accordingly, we have carefully revised the manuscript by taking all comments and suggestions into account in the revision. Our point-by-point responses to these comments are summarized as follows. Our response is marked with "Blue". Meanwhile, we give the tracked-changes version of the manuscript. Note that the **Line numbers** and Fig./Table numbers in this response are **corresponded to the track-changed version** of the revised manuscript.

REVIEWER COMMENTS

Reviewer #1 (Remarks to the Author):

Review of Manuscript "A New Climate Oscillation Takes Typhoon Activity Toward Asian High-Latitude Cities"

This manuscript presents a novel climate oscillation, the tropical-leaning North Atlantic Oscillation (TNAO), and explores its predictive capability for the poleward migration of typhoon activity in East Asian high-latitude cities. The study makes a significant contribution to the field by identifying a distinct oscillation from the classical North Atlantic Oscillation (NAO) and demonstrating its robust predictive skill for typhoon tracks. The findings are particularly relevant for improving seasonal forecasts and disaster preparedness in the region under the background of global climate change.

The identification of the TNAO as a distinct oscillation from the NAO is a major contribution. The results and conclusions in study are basically convincing. Also, the use of multiple datasets (e.g., ERA5, CMIP6) and statistical techniques (e.g., EOF analysis, wave activity flux) strengthens the credibility of the findings. The exclusion

of extreme El Niño years enhances the robustness of the TNAO-TTD relationship. This study clearly links TNAO to jet stream variability, SST anomalies (PMM, NAT), and subsequent typhoon steering flows. The cross-basin teleconnection mechanism is well-articulated.

Generally, this study makes a compelling case for the TNAO as a key driver of typhoon variability. With minor revisions to clarify mechanisms and contextualize findings, it would be a valuable contribution to high-impact journals. The reviewer recommend acceptance after addressing the above points.

Response: We sincerely appreciate the reviewer’s positive and encouraging comments. We are grateful for the constructive suggestions for improvement. We have carefully revised the manuscript to further clarify the physical mechanisms and enhance the discussion of the results as suggested. Please check our response below.

Major comments:

(1) While the distinction from NAO is emphasized, further discussion is needed on why TNAO was not identified earlier. A spectral analysis or comparison with other known modes (e.g., Atlantic Multidecadal Oscillation) could solidify its uniqueness.

Response: The reviewer's suggestions were very meaningful. We discussed them accordingly and added TNAO and wavelet coherence analysis of multiple modalities (**Fig. R1.1**), as follows:

Lines 76–84: “There are several points need to be emphasized: First, due to the great fame and wide-spread climate impacts of the extra-tropical NAO, it stealthily shields other tropical–extra-tropical atmospheric oscillation. In addition, the TNAO appears only during cold seasons (Supplementary Fig. 2) when the NAO is most robust, but substantially weakens and even disappears during warm seasons when NAO is also dominant. In other words, only when the NAO signal is removed, can the cross-latitudinal correlation of sea level pressure (SLP) directly capture the previously-overlooked TNAO signal (Fig. 1a). This may be the reason why the

TNAO was not identified earlier.”

Lines 135–137: “Besides, a comparison with other known decadal modes (e.g., AMO and PDO) via the wavelet coherence spectral analysis shows that the TNAO is indeed distinct from the NAO (Supplementary Fig. 4), further solidifying its uniqueness.”

Fig. R1.1 (i.e., Supplementary Fig. 4). Wavelet coherence analysis of multiple modes. Wavelet coherence spectral analysis **a** between TNAO index and AMO index, **b** between TNAO index and PDO index, **c** between NAO index and AMO index, and **d** between NAO index and PDO index, respectively. Contours denote passing the 95% confidence level. Vectors denote the phase relationship of two indices.

(2) Please clarify whether TNAO is independent of ENSO on longer timescales, as only extreme El Niño years were excluded.

Response: We appreciate the reviewer’s insightful comment. Indeed, examining

the potential influence of ENSO is important. Therefore, we provided scatter plots between Niño3.4 and TNAO under different conditions (**Fig. R1.2**). On longer timescale of 1958–2023, TNAO is still independent of ENSO (**Fig. R1.2a–b**). Similarly, during 1979–2023, relationship between Niño3.4 and TNAO show weak correlations (**Fig. R1.2c–f**). After removing ENSO years, the correlation between TNAO and TTD remains significant ($R = 0.72, p < 0.01$; **Fig. R1.2f**). These results suggest that our main conclusion of the TNAO–TTD correlation is largely independent of ENSO. Combined with the comments of other reviewers on ENSO, we make the following general modifications:

Lines 149–153: “Given that relationship between TNAO and ENSO is not significant (even in a longer period of 1958–2023; Supplementary Fig. 6a–e), we have tested and found that even in the case of excluding all ENSO years, the TNAO-TTD relationship still keeps robust ($R = 0.72, p < 0.01$; Supplementary Fig. 6f–g). Accordingly, the cross-seasonal TNAO-TTD linkages are generally independent of ENSO.”

Fig. R1.2 (i.e. part of Supplementary Fig. 6). Relationship between ENSO and TNAO. Scatter plot of spring TNAO versus pre-winter (a) and peak-season (b)

Niño3.4 indices during 1958–2023. **c–d** Same as **a–b** but during 1979–2023. **e** Same as **d** but excluding 1982, 1997 and 2015. **e** Same as **d** but further excluding extreme La Niña years of 1988, 1999 and 2010 (selected by three lowest Niño3.4 index years). **f** Scatter plot of spring TNAO versus TTD excluding El Niño and La Niña years that Niño3.4 index exceeding 1.0 absolute value. Correlation coefficients and their p-values are marked.

(3) The Gill-type Rossby wave response and "subtropical Atlantic-Pacific teleconnection" require more detailed explanation or references. A schematic diagram could aid understanding.

Response: Here we added detailed explanation (Fig. R1.3-R1.4) and a schematic diagram (Fig. R1.5). Please check the following mechanism:

Lines 197–207: “Due to the TNAO-promoted summer tropical SST changes, significant atmospheric response can be induced over the WNP, especially the lower-level WNP cyclonic circulation anomaly (streamlines in Fig. 3a). Moreover, positive PMM linked to TNAO can promote active convection over the subtropical North Pacific (Supplementary Fig. 11), inducing a Rossby wave cyclone over the WNP. Moreover, the lower-level WNP cyclonic circulation can be also identified as the WNP branch of the Pacific–Atlantic subtropical oscillation (PASO)⁴² driven by the TNA SST anomalies via inducing subtropical Walker-cell-like changes (Supplementary Fig. 12) and Pacific convection response (Supplementary Figs. 11b and d), which further contribute to WNP cyclone^{44,45}.”

Lines 370: “A schematic diagram is presented in Fig. 6.”

Fig. R1.3 (i.e., Supplementary Fig. 11). TNAO-associated atmospheric anomaly. **a** Regression of 850-hPa horizontal wind (vector) and correlation of precipitation (shading) during March–May onto TNAO index. Red rectangle is the same as in Supplementary Fig. 9. **b** Same as **a** but variables are during July–September. **c–d** Same as **a–b** but for 200-hPa horizontal wind and outgoing longwave radiation. All the shadings and vectors are significant at the 95% confidence level.

Fig. R1.4 (i.e., Supplementary Fig. 12). TNA-associated atmospheric anomaly. a Time series of TNA (blue) and PASO (red) during July–September from 1982 to 2023. Correlation between them are marked. Note that after removing the long-term trend, correlation coefficient remains as high as 0.66 ($p < 0.01$). **b** Regression of SLP onto inverse TNA (contour; interval: 0.2 hPa) and inverse PASO (shading). **c** Regression of vertical circulation (streamline) and corresponding vertical velocity (shading) between 10°N–30°N onto reverse TNA index during July–September. Hatching indicates shading passes the 95% confidence level.

Fig. R1.5 (i.e., Fig. 6). Schematic diagram illustrating the influence of the TNAO on WNP typhoons. During spring, the TNAO weakens trade winds over the subtropical northeastern Pacific, leading to local SST warming and subsequent development of the PMM through the seasonal footprint mechanism. Concurrently, TNAO strengthens the trade winds over the TNA, resulting in SST cooling that persists into summer via local air-sea coupling. The cold TNA SST anomalies can continuously induce Rossby wave anticyclone to its west, thereby further promoting PMM development. The Atlantic-eastern Pacific Rossby wave anticyclone has been demonstrated to directly favor the WNP cyclone via Pacific convection, a phenomenon that is explained by the PASO. Consequently, the presence of a lower-level cyclone and an upper-level anticyclone in the WNP leads to an eastward shift in the TUTT and MT (solid line for positive TNAO and dashed line for negative TNAO), promoting more (less) frequent typhoon genesis in the eastern (western) WNP. The altered steering flow favors (inhibits) typhoon movement towards higher (lower) latitudes of East Asia.

(4) please address how TNAO-induced SST anomalies persist into summer. Is this due to ocean memory or atmospheric feedback?

Response: TNAO-induced SST anomalies can continue to develop (PMM) and maintain (NAT) through coupling with the atmosphere. We supplemented the changes

in the climate state wind field (**Fig. R1.6**) to correspond to the changes in the SST mode (**Fig. R1.7**) and explained the mechanism in more detail:

Lines 179–189: “The anticyclonic flows can weaken local trade winds and warm SST in the subtropical eastern North Pacific (red rectangle in Supplementary Fig. 9), which subsequently drives PMM development via the seasonal footprint mechanism^{25,33}. On the other hand, the TNAO can induce the NAT SST anomalies (Supplementary Fig. 8a). Specially, the TNAO-related anomalous northeasterly winds over the tropical North Atlantic (TNA) can enhance the trade winds to cool SST there (blue rectangle in Supplementary Fig. 9). This cold SST can further strengthen the anticyclone over the eastern Pacific–North Atlantic via the Gill-type Rossby wave response^{34,35}. Therefore, this cold SST can persist into the following summer (Supplementary Fig. 8c) and further contribute to the development of PMM (Supplementary Fig. 8d) through a Gill-type Rossby wave response with an anticyclone to its west^{33,36–38}.”

Fig. R1.6 (i.e., Supplementary Fig. 9). Climatology of SST and 850-hPa horizontal wind over the Pacific and the Atlantic during March–May. Red (blue) vectors indicate that climatic wind will enhance (weaken) associated with positive TNAO (diagnosed by climatic wind speed compared with wind speed plus the regression onto TNAO index). Black contour in the Pacific is the 0.5 correlation coefficient of SST onto the July–September PMM index, representing the origin of PMM. Red and blue

rectangles show the regions of PMM development and NAT maintenance, respectively.

Fig. R1.7 (i.e., Supplementary Fig. 8). SST evolution related to TNAO. a Correlation pattern of SST in April–June (shading) and 850-hPa horizontal wind in March–May (vector) onto the spring TNAO index. **b** Lead-lag correlation of seasonal PMM onto the spring TNAO index. The horizontal dashed line denotes the threshold of the 95% confidence level. **c–d** Same as **b** but for seasonal NAT and spring TNAO index, and seasonal PMM and April–June NAT index, respectively.

(5) about the CMIP6 Projections: the 39.4% increase in positive TNAO events is intriguing but lacks discussion on model spread or uncertainties. How do model biases (e.g., in Atlantic SST) affect this projection?

Response: We agree that the uncertainty of model results is vital. We have discussed the cold biases in Atlantic SST, which may indeed introduce some

uncertainty into future projections for the TNAO. However, current model results show high consistency, leading us to conclude that the reliability of future TNAO projections remains strong. The relevant modifications are as follows:

Lines 343–353: “While we suggest that CMIP6 models show a robust projection of more frequent positive TNAO events, some uncertainties remain. The spread among models primarily arises from differences in simulating the mean-state and variability of North Atlantic SSTs, as well as the subtropical jet response^{61,62}. It is reported that most CMIP6 models exhibit cold SST biases in the North Atlantic, especially in the tropics^{63,64}. This may cause an overestimation of positive TNAO frequency because positive TNAO corresponds to negative TNA SST anomaly. Nevertheless, the overall positive shift in TNAO occurrence is consistent across most models, indicating that the projected intensification of positive TNAO is a robust multi-model signal rather than an artifact of a few outliers. Future work could further assess the sensitivity of TNAO projections to Atlantic SST biases and internal variability through targeted model experiments.”

Other comments and suggestions:

(1) the statement on decelerated typhoon motion seems tangential; either expand or omit.

Response: Done as suggestion. We intend to highlight that the slowing movement speed of typhoons can lead to an extended duration of their impact, thereby underscoring the growing threat posed by typhoons moving northward in the future. We added more explanation as follows:

Lines 296–300: “In addition, the future anthropogenic warming would decelerate typhoon motions near high-latitude populated coastal regions due to a poleward shift of the subtropical westerlies, **which can induce a longer time of typhoon impacts**⁵⁵. And accompanying with increasing intensity and rainfall⁵⁶, this can potentially compounds future typhoon-related damages.”

(2) The abstract overstates "inevitably face more serious threats." Modify to reflect uncertainties in CMIP6 projections.

Response: Done as suggestion. We revised the sentence as follows:

Lines 33–36: “Climate models project an increasing frequency of positive TNAO events. This implies that East Asian high-latitude cities would potentially face more serious typhoon threats as climate warms, yet uncertainty remains due to model biases in simulating Atlantic sea surface temperature.”

(3) Discuss how TNAO’s predictive skill compares to existing indices (e.g., PDO, AMO).

Response: The reviewer's suggestions warrant careful analysis. We have compiled correlation coefficients between key SST modes during the winter and spring seasons and the TNAO and TTD (**Table R2.1**). As shown, none of these SST modes exhibit significant linear correlations with either the TNAO or TTD. This indicates that the influence of TNAO on TTD operates independently of these SST modes, highlighting the unique predictive role of TNAO. The corresponding modifications are as follows:

Lines 285–293: “In addition to the above climate variables, SST variability often has a significant influence on typhoons. However, unlike metrics such as typhoon genesis location, frequency or intensity⁴⁷⁻⁵², the relationship between other SST variability and TTD remains unexplored. Therefore, we further calculate the correlation coefficients between the winter-spring SST indices and the TNAO as well as the TTD. As shown in Supplementary Table 2, there was no significant linear correlation between TNAO (or TTD) and other existing indices (Niño3.4, PDO, AMO and Indian Ocean Dipole). This suggests that the influence of the TNAO on the TTD is generally independent of these SST modes. These findings also highlight the unique predictive significance of the TNAO for the northward shift of East Asian typhoons.”

Table R1.1 (i.e., Supplementary Table 2). Relationship between preceding winter/spring SST modes and TNAO/TTD. Correlation coefficients of preceding winter (December–January–February, DJF) and spring (March–April–May, MAM) Niño3.4, AMO, PDO, Indian Ocean Dipole (IOD) with spring TNAO and peak season (July–August–September, JAS) TTD during 1979–2023. None of these correlations is significant at the 90% confidence level of Student’s *t*-test.

Season of SST indices	R	Niño3.4	AMO	PDO	IOD
DJF	TNAO_MAM	−0.19	0.15	0.01	−0.16
	TTD_JAS	−0.12	0.03	0.00	−0.05
MAM	TNAO_MAM	−0.12	−0.18	0.19	−0.18
	TTD_JAS	−0.04	−0.13	0.11	−0.08

(4) Highlight limitations, such as the assumption of stationary TNAO-typhoon relationships under climate change.

Response: Done as suggestion. Discussion regarding possible change TNAO-TTD relationship is added:

Lines 326–335: “It should be noted that our hypothesis is based on the assumption that the relationship between the TNAO and TTD remains unchanged. Additionally, we suggest that the TNAO-TTD relationship is unchanged after excluding the long-term trends (Supplementary Figs. 14–15). However, under the context of global warming, typhoon activity is evolving. For instance, the frequency of future typhoons may continue to decrease with global warming⁵⁷⁻⁶⁰, which could alter the response of typhoon to TNAO forcing. On the other hand, the influence of TNAO on typhoons depends on teleconnection processes between the ocean and atmosphere, which may also change with global warming, thereby altering the TNAO-TTD relationship. Therefore, the conjecture presented in this study remains highly preliminary, and further research is needed.”

Reviewer #2 (Remarks to the Author):

Comments on “A new climate oscillation takes typhoon activity toward Asian high-latitude cities” by Wu et al.

This study focuses an important issue of atmospheric mode on NWP typhoons. In general, this manuscript is well written and shows a clear story line of how TNAO impacts NWP typhoons. However, I really concern the novelty and robustness, which could need extra analyses.

Response: We sincerely thank the reviewer for the insightful comments on our manuscript. Accordingly, we have conducted additional analysis to highlight the novelty and robustness of this study, which can substantially improve the quality of the manuscript. Please check our point-to-point response below.

Major Comments:

1. The authors show the TNAO impact on tropical cyclone over the NWP. However, I find that the pathway TNAO impact is much similar to the North Tropical Atlantic (NTA) SST impact on Pacific climate (Ham et al. 2013). In this sense, I considered that the TNAO and NTA SST are air-sea coupled system that they share the same physical process affecting the NWP Typhoon. Moreover, the TNAO is relatively transient compared with the NTA and the statistical covariance is fragile. Therefore, the verification of TNAO impact on NWP typhoon activity in numerical models is of great importance.

Reference:

Ham et al. Sea surface temperature in the north tropical Atlantic as a trigger for El Niño/Southern Oscillation events

Response: The reviewer’s perspective is highly significant, and we also agree that the tropical North Atlantic (TNA) SST and TNAO share common physical processes. Just as proposed by Ham et al. (2013), negative TNA SST anomalies first induce an anomalous Rossby anticyclone in its western flank, which subsequently

influences the development of North Pacific PMM. This process has been demonstrated in their numerical experiments.

Our mechanism shares similarities: the positive TNAO-related anticyclone (i.e., the southern part of TNAO) triggers the PMM development in the North Pacific since spring, whereas the spring TNAO-induced TNA SST anomalies persist to summer and thereby influence North Pacific climate in the following seasons (Fig. 3a). In other words, this indicates that the TNAO exerts a more direct impact on the North Pacific climate, suggesting it may be a superior and earlier typhoon TTD forecasting factor compared to the TNA.

Nevertheless, we also compare it with the influence of TNA on typhoons. As shown in **Figs. R2.1a-b**, the TNA-related typhoon anomalies are markedly weaker than those associated with TNAO. Although TNA also leads to late-stage PMM-like SST development (**Figs. R2.1c-d**), the associated cyclonic anomalies in the WNP are much weaker than those driven by TNAO (compared to **Fig. 3c**). However, the reviewer's perspective provided a valuable insight: Given that the TNA influence mechanism has been supported by numerical models (Ham et al. 2013), the TNAO impact on Pacific climate is likewise reliable. In other words, the numerical experiments of Ham et al. (2013) can support our mechanism.

On the other hand, although the TNAO itself is transient, its effects on the TNA operate through the ocean memory, which can develop and persist into the typhoon peak-season. Thus, the short-lived nature of TNAO on its own timescale does not affect its relationship with typhoons. Accordingly, we have revised the mechanism in details (**Figs. R2.2–2.3**) and added related discussion as follows:

Lines 176–196: “Mechanisms of TNAO performance in predicting TTD-like typhoon activity can be explained as follows. On the one hand, the TNAO can promote the Pacific Meridional Mode (PMM) SST anomalies through the anticyclonic flows over the northeastern Pacific (Supplementary Figs. 8a–b). The anticyclonic flows can weaken local trade winds and warm SST in the subtropical eastern North

Pacific (red rectangle in Supplementary Fig. 9), which subsequently drives PMM development via the seasonal footprint mechanism^{25,33}. On the other hand, the TNAO can induce the NAT SST anomalies (Supplementary Fig. 8a). Specially, the TNAO-related anomalous northeasterly winds over the tropical North Atlantic (TNA) can enhance the trade winds to cool SST there (blue rectangle in Supplementary Fig. 9). This cold SST can further strengthen the anticyclone over the eastern Pacific–North Atlantic via the Gill-type Rossby wave response^{34,35}. Therefore, this cold SST can persist into the following summer (Supplementary Fig. 8c) and further contribute to the development of PMM (Supplementary Fig. 8d) through a Gill-type Rossby wave response with an anticyclone to its west^{33,36-38}. And the physical process from TNA SST to PMM has been demonstrated by Ham et al.³⁷ through observations and numerical experiments. However, the impact of TNAO on TTD is much more pronounced than that of TNA SST (Supplementary Fig. 10), where only the northward-moving typhoon is significantly altered^{40,41}. This may be because that TNAO not only affects Pacific climate through the persistent TNA SST anomalies in later seasons but also directly triggers the formation of the early prototype of PMM via the southern part anticyclonic circulation of TNAO during spring.”

Fig. R2.1 (i.e., Supplementary Fig. 10). TNA SST and typhoon activity. a Regressions of peak season typhoon genesis density onto the spring Tropical North Atlantic (TNA) SST index during 1979–2023. **b** Same as **a** but for track density. **c** Correlation of SST (shading) and regression of 850-hPa horizontal wind (vector; $p < 0.05$) during spring (March–April–May) onto the spring TNA index (mean SST in 80°W – 20°E , 0° – 20°N). **d** Same as **c** but SST and horizontal wind are during peak season (July–August–September).

Fig. R2.2. (i.e., Supplementary Fig. 8). SST evolution related to TNAO. a Correlation pattern of SST in April–June (shading) and 850-hPa horizontal wind in March–May (vector) onto the spring TNAO index. **b** Lead-lag correlation of seasonal PMM onto the spring TNAO index. The horizontal dashed line denotes the threshold of the 95% confidence level. **c–d** Same as **b** but for seasonal NAT and spring TNAO index, and seasonal PMM and April–June NAT index, respectively.

Fig. R2.3. (i.e., Supplementary Fig. 9). Climatology of SST and 850-hPa horizontal wind over the Pacific and the Atlantic during March–May. Red (blue) vectors indicate that climatic wind will enhance (weaken) associated with positive TNAO (diagnosed by climatic wind speed compared with wind speed plus the regression onto TNAO index). Black contour in the Pacific is the 0.5 correlation coefficient of SST onto the July–September PMM index, representing the origin of PMM. Red and blue rectangles show the regions of PMM development and NAT maintenance, respectively.

2. The authors considered the poleward migration of NWP typhoon under global warming. However, many studies have shown that the NWP typhoon frequency will decrease under the global warming (Knutson et al. 2020). The CMIP6 models can only reflect the TNAO changes, however, the connections between NWP typhoons and TNAO may also change. The changed NWP typhoons may also shape the NWP typhoon – TNAO relationship.

Reference:

Knuston et al. Tropical Cyclones and Climate Change Assessment: Part II: Projected Response to Anthropogenic Warming

Response: We fully agree with the reviewer's perspective. This point had previously escaped our consideration, so we have revised our original statement to discuss the potential changes in the relationship between TNAO and typhoons and present our speculations with greater caution. The relevant revisions are as follows:

Lines 322–331: “It should be noted that our inference is based on the premise that the linkage between the TNAO and TTD remains unchanged. Since the cross-seasonal linkage between the two keeps unchanged even after excluding the long-term trends (Supplementary Figs. 15–16). However, under the context of global warming, typhoon activity is evolving. For instance, the frequency of future typhoon may continue to decrease with global warming⁵⁹⁻⁶², which could alter the response of typhoons to TNAO forcing. On the other hand, the influence of TNAO on typhoons

depends on teleconnection processes between the ocean and atmosphere, which may also change with global warming, thereby altering the TNAO-TTD relationship. Therefore, the conjecture presented in this study remains highly preliminary, and further research is needed.”

Fig. R2.4 (i.e., Supplementary Fig. 15). TNAO modulations on typhoon and circulation. Same as Fig. 2a–b, 3a–c, but trend is removed. a Scatter diagram of the spring TNAO (red) and extratropical NAO (blue) against the TTD of peak season typhoon track density. Correlation coefficient (R) between the TNAO (extratropical

NAO) and TTD is marked. Straight Lines –are linear least-square fits. **b** Regressions of peak season typhoon track density onto the TNAO (shading) and TTD (contours) indices. **c** Shading in the Pacific (Atlantic) is the correlation pattern of SST in July–September onto the simultaneous PMM index (April–June NAT index). StreamLines –are the regressed 850-hPa horizontal wind onto the PMM+NAT index (WNP region). Contours indicate three atmospheric systems: MT (represented by the relative vorticity of $3 \times 10^{-6} \text{ s}^{-1}$ at 850 hPa), TUTT (represented by the 12420-gpm geopotential height at 200 hPa) and South Asian high (SAH; represented by the 12500-gpm geopotential height at 200 hPa). Solid (dashed) contours are calculated as the climatology plus (minus) the double regressions onto the PMM+NAT index. **d** Regressions of peak season Genesis Potential Index (shading) and typhoon genesis density (contour; 10^{-1} yr^{-1}) on the spring TNAO index. **e** Regressions of peak season 700-hPa geopotential height (shading) and steering flow (vector; $p < 0.05$) onto the spring TNAO index. Hatching indicates that it passes the 95% confidence level.

Fig. R2.5 (i.e., Supplementary Fig. 16). SST evolution related to TNAO. Same as Supplementary Fig. 5, but trend is removed. **a** Correlation pattern of SST in April–June (shading) and 850-hPa horizontal wind in March–May (vector) onto the spring TNAO index. **b** Lead-lag correlation of seasonal PMM onto the spring TNAO index. The horizontal dashed line denotes the threshold of the 95% confidence level. **c–d** Same as **b** but for seasonal NAT and spring TNAO index, and seasonal PMM and April–June NAT index, respectively.

3. The authors only considered the typhoons, which reach 32.6m/s and above, right? However, I believe that the TNAO could affect both typhoons and tropical cyclones (17-32.6 m/s) in the NWP. The authors need to show the TNAO impact on all TC frequency and explain why they only focused on typhoons?

Response: We appreciate the reviewer’s comment. Following the suggestion, we have examined the relationship between the TNAO and the frequency of all tropical cyclones (TCs, $\geq 17.2 \text{ m s}^{-1}$) in the Northwest Pacific (**Fig. R2.6**). The result confirms that the TNAO significantly modulates overall TC East Asian dipole mode, with a correlation of $R = 0.60$. This is consistent with the strong influence found for typhoons ($\geq 32.6 \text{ m s}^{-1}$), for which the correlation is higher ($R = 0.75$). These analyses demonstrate that the TNAO exerts robust impacts across the full spectrum of TC intensities.

Our focus on typhoons was motivated by their disproportionately large societal and climatic impacts compared with weaker TCs, including more severe damages, economic losses, and fatalities (e.g., Knutson et al., 2020; Zhai et al., 2014;). Therefore, highlighting typhoons emphasizes the relevance of the TNAO to high-impact events. Nevertheless, we agree that the TNAO is also a useful predictor of overall TC variability, and we now clarify this point in the revised manuscript as follows:

Lines 390–394: “We focus on the East Asia–WNP typhoon (TC with $V_{\max} > 32.6$

m s⁻¹) during July–September of 1979–2023, using the Joint Typhoon Warning Center TC best-track datasets. Although TNAO exerts some influence on typhoon activity in other months, its relationship with the TTD mode is mainly concentrated in the peak season (i.e., July–September) when WNP typhoon most active (Supplementary Fig. 22). Thus we mainly focus on July–September. This study primarily focuses on typhoons because stronger typhoons are more likely to cause significant damage^{60,75}. We also examined results for all TCs with $V_{\max} > 17.2$ m s⁻¹, which showed consistent findings (Supplementary Fig. 23), although the relationship with TNAO may be somewhat weaker for weaker TCs due to their inherent randomness.”

Fig. R2.6 (i.e., Supplementary Fig. 22). Relationship between TNAO and all TC activity during peak season. Same as Fig.2a–b but for TCs with a $V_{\max} > 17.2$ m s⁻¹.

References:

- Knutson, T., Camargo, S. J., Chan, J. C., Emanuel, K., Ho, C. H., Kossin, J., ... & Wu, L. (2020). Tropical cyclones and climate change assessment: Part II: Projected response to anthropogenic warming. *Bulletin of the American Meteorological Society*, 101(3), E303-E322. <https://doi.org/10.1175/BAMS-D-18-0194.1>
- Zhai, A. R., & Jiang, J. H. (2014). Dependence of US hurricane economic loss on maximum wind speed and storm size. *Environmental Research Letters*, 9(6),

Minor comments:

1. Line 299: The definition of jet shift index could be incorrect. Please modify the definition region. Also, it is worthy noted that the authors defined too many atmospheric modes in the Method part, I suggested to revise the definitions and explain the purpose of the usage.

Response: We appreciate the reviewer for pointing out this issue: the original definition of jet shift index was indeed not reasonable enough. We have revised this index by expanding its definition regions (as shown by the boxes in **Fig. R2.7a**). This modification allows the jet shift index to capture the meridional variation of the jet axis (**Fig. R2.7b**) and slightly improving the correlation coefficient with the TNAO (from 0.83 to 0.85).

Furthermore, the original manuscript contained multiple index definitions, which may have appeared confusing. To present the definitions more clearly, we have consolidated them into the **Table R2.1** and revised the manuscript as follows:

Lines 115–117: “Moreover, the TNAO index has a robust Pearson correlation coefficient up to 0.85 ($p < 0.01$) with the North America-Atlantic jet meridional shift index (NAJS; see Methods, Fig. 1c and Supplementary Fig. 3).”

Lines 420–430: “This paper employs multiple indices, primarily atmospheric modes and SST indices, with specific definitions provided in Supplementary Table 3. We define the spring (March–May) TNAO and NAO indices based on the difference in SLP between two regions. We compare our NAO index with downloaded NAO index from <https://climatedataguide.ucar.edu/climate-data/hurrell-north-atlantic-oscillation-NAO-index-pc-based>, which is the leading EOF mode of SLP in (90°W–40°E, 20°N–80°N), and find that two time series show strong Pearson correlation ($R = 0.97$, $p < 0.001$) during spring. The SST index is defined as the regional average sea surface temperature. We utilized multiple atmospheric or high-latitude modal indices from

preceding winter and spring seasons (with specific seasons determined by high correlation coefficients in Supplementary Fig. 25) to highlight the unique and dominant role of TNAO.”

Fig. R2.7 (i.e., Supplementary Fig. 3). Definition of the jet shift index. a Difference in 200-hPa zonal wind between positive and negative phases of the jet meridional shift index, defined based on one standard deviation. Positive years are 1979, 1980, 1981, 1983, 1984, 1987, 1988, 1998, 1999, 2005, 2006, 2010, and 2013. Negative years are 1985, 1989, 1991, 1994, 2000, 2002, 2004, 2009, 2018, 2019, and 2022. **b** Composite of 200-hPa zonal wind speed during positive years (contours and thick blue line) and negative years (shading and thick red line). The thick Lines – indicate the jet axis, defined as the zero line of the meridional gradient of zonal wind. **c** Time series of TNAO (red), TTD (blue) and NAJS (gray) and correlation coefficients between them.

Table R2.1 (i.e. Supplementary Table 3). Definition of indices used in this manuscript. In this study, we use various indices during 1979–2023, including the TNAO, NAO, NAT, PMM, TNA, PASO, NAJS, Northern Annular Mode (NAM), North Pacific Oscillation (NPO), and Arctic sea-ice concentration (SIC). Seasons are abbreviated by the first letter of the month. All the subscripts denote the regionally or zonally averaged and standardized variables.

Index	Variable (Season)	Definition
TNAO	Sea level pressure (MAM)	$SLP_{(25^{\circ}W-105^{\circ}W, 15^{\circ}N-35^{\circ}N)} - SLP_{(45^{\circ}W-95^{\circ}W, 55^{\circ}N-65^{\circ}N)}$
NAO	Sea level pressure (MAM)	$SLP_{(60^{\circ}W-30^{\circ}E, 40^{\circ}N)} - SLP_{(60^{\circ}W-30^{\circ}E, 70^{\circ}N)}$
NAT	Sea surface temperature (AMJ)	$SST_{(30^{\circ}W-50^{\circ}W, 45^{\circ}-55^{\circ}N)} - SST_{(40^{\circ}W-100^{\circ}W, 25^{\circ}N-40^{\circ}N)} - SST_{(10^{\circ}W-60^{\circ}W, 7.5^{\circ}N-22.5^{\circ}N)}$
PMM	Sea surface temperature (JAS)	$SST_{(160^{\circ}E-175^{\circ}W, 8^{\circ}N-18^{\circ}N)} + SST_{(110^{\circ}W-130^{\circ}W, 14^{\circ}N-20^{\circ}N)}$
TNA	Sea surface temperature (JAS)	$SST_{(80^{\circ}W-20^{\circ}E, 0^{\circ}-20^{\circ}N)}$
PASO	Sea level pressure (JAS)	First leading principal component of $SLP_{(110^{\circ}E-30^{\circ}W, 10^{\circ}N-30^{\circ}N)}$
NAJS	200-hPa zonal wind (MAM)	$U200_{(40^{\circ}W-105^{\circ}W, 42.5^{\circ}N-52.5^{\circ}N)} - U200_{(40^{\circ}W-105^{\circ}W, 22.5^{\circ}N-32.5^{\circ}N)}$
NAM	Sea level pressure (MAM)	$SLP_{(0^{\circ}-360^{\circ}, 45^{\circ}N)} - SLP_{(0^{\circ}-360^{\circ}, 80^{\circ}N)}$
NPO	Sea level pressure (MAM)	$SLP_{(160^{\circ}E-140^{\circ}W, 30^{\circ}N)} - SLP_{(170^{\circ}E-120^{\circ}W, 65^{\circ}N)}$
SIC	Sea ice concentration (FMA)	$SIC_{(160^{\circ}W-180^{\circ}, 60^{\circ}N-68^{\circ}N)}$

2. It is unfair to move the TNAO signal from the NAO index since they are highly correlated in Fig.4. The authors could also check the TNAO-TC track correlation when removing the NAO signal. I believe it is also weak and insignificant.

Response: Done as suggestion. Our results indicate that even after linearly removing NAO signal, TNAO shows high connection with the TTD pattern (Fig.

R2.8). NAO has a high correlation with typhoons that move northward toward northern East Asia (**Fig. R2.9**), therefore removing NAO signal weakens this signal in TNAO-related typhoon track pattern (positive correlation in **Fig. R2.8d**). Still, correlation pattern of typhoon track density onto TNAO significantly exhibits the meridional dipole, confirming the advance and robustness of TNAO in predicting TTD than that of NAO. We revised the manuscript as follows:

Lines 271–280: “We further compare the correlation pattern of Prediction_{TTD} (Fig. 4b) with other prediction models established only using TNAO (Fig. 4c–d) and NAO (Fig. 4e and Supplementary Fig. 19). The Prediction_{TTD} explains 61.7% of the observed variances in TTD variations during 1979–2023 (evaluated by the R^2 between indices; Fig. 4b), while the TNAO alone explains a substantial 56.3% of the variation (Fig. 4c). When the NAO signal is removed, TNAO maintains its strong linkage to the TTD pattern (Fig. 4d). In contrast, the extratropical NAO-model (Supplementary Fig. 19) reflects very low predictability ($R^2 = 14.7\%$). Moreover, when the TNAO signal is removed, the predicted variances from the extratropical NAO decreases to approximately 0% (Fig. 4e), suggesting that the relationship between the extratropical NAO and TTD is not independent of the TNAO.”

Fig. R2.8 (i.e., Fig. 4). Predicting Asia–Pacific typhoon activity. **a** Time series of standardized observed TTD (blue bar) and predicted TTD (red line) during 1979–2023. Prediction is divided into training period (1979–2014; blue shading) and validation period (2016–2023; red shading). The correlation and significance level between observation and prediction are marked, also the RMSE for two periods. **b–e** Correlation pattern between typhoon track density and time series predicted by different predictors (see Methods for details) in 1979–2023: using TNAO, extratropical NAO, NAM, NPO and SIC (**b**), TNAO only (**c**), TNAO only and excluding NAO signal (**d**), NAO only and excluding TNAO signal (**e**). Hatching indicates that the correlation passes the 95% confidence level. The corresponding R^2 between TTD index and prediction model is marked.

Fig. R2.9 (i.e., Supplementary Fig. 19). Same as Fig. R1.4b, but for NAO.

3. Since the TNAO is induced by the jet shear, I also want to the correlations between North America jet stream and NWP typhoons (and TNAO).

Response: Done as suggestion. In Fig. R2.7, we also show the seasonal correlation between the North American jet stream and TNAO and TTD. In spring, the North American jet stream has the highest correlation with TNAO and peak season TTD, which are 0.85 and 0.64, respectively (both $p < 0.01$). We added this information in the manuscript:

Lines 144–145: “Besides, the NAS has a weaker correlation of 0.64 with TNAO (Supplementary Fig. 3c).”

4. Line 126: why do they authors only remove three strong El Niño events? I believe that the TNAO is highly correlated with the NTA SST simultaneously and consequently ENSO signal. Therefore, it may also need to consider both El Niño and La Niña events.

Response: We appreciate the reviewer’s insightful comment. Indeed, examining the potential influence of ENSO is important. Therefore, we provided scatter plots between Niño3.4, TNAO, and TTD under different conditions (Fig. R2.10). During 1979–2023, Niño3.4 and TNAO show a weak correlation ($R = 0.28$, $p = 0.07$; Fig.

R2.10a). After removing three strong El Niño years (1982, 1997, 2015), the correlation becomes insignificant ($R = 0.18$, $p = 0.25$; **Fig. R2.10b**). When we instead remove the three extreme El Niño and La Niña years (1982, 1997, 2015, 1988, 1999, and 2007), the correlation of Niño3.4 and TNAO keeps insignificant (**Fig. R2.10c**), but the TNAO–TTD relationship keeps robust ($R = 0.73$; **Fig. R2.10d**), just slightly weakens compared to that in the main text ($R = 0.75$). Furthermore, excluding all El Niño and La Niña years (i.e., 1982, 1987, 1988, 1997, 1998, 1999, 2002, 2007, 2010, 2015, 2022 and 2023) yields a similar result ($R = 0.72$; **Fig. R2.10e**), albeit slightly lower than 0.75. These results confirm that our main conclusion is robust, ensuring that the TNAO–TTD correlation remains as high as possible while maintaining an insignificant linear relationship between TNAO and Niño3.4. We also examine the multiple regression when excluding both extreme El Niño and La Niña years (**Fig. R2.11**), and found that results are also robust.

Lines 149–153: “Given that relationship between TNAO and ENSO is not significant (even in a longer period of 1958–2023; Supplementary Fig. 6a–e), we have tested and found that even in the case of excluding all ENSO years, the TNAO-TTD relationship still keeps robust ($R = 0.72$, $p < 0.01$; Supplementary Fig. 6f–g). Accordingly, the cross-seasonal TNAO-TTD linkages are generally independent of ENSO signal.”

Lines 267–268: “This prediction is generally independent of ENSO (Supplementary Fig. 18).”

Fig. R2.10 (i.e., part of Supplementary Fig. 6). Relationship between ENSO and TNAO. Scatter plot of spring TNAO versus peak-season (a) Niño3.4 index during 1979–2023. (b) Same as a but excluding years 1982, 1997 and 2015. (c) Same as b but further excluding extreme La Niña years of 1988, 1999 and 2010 (selected by three lowest Niño3.4 index years). (d) Same as c but for TNAO versus TTD. (e) Scatter plot of spring TNAO versus TTD excluding El Niño and La Niña years that Niño3.4 index exceeding 1.0 absolute value. Correlation coefficients and their p-values are marked.

Fig. R2.11 (i.e., Supplementary Fig. 18). Same as Fig. 4a, but further excluding three extreme La Niña years (1988, 1999 and 2010).

5. Line 273 and Line 73: the typhoon season is defined as the months from July to September or August? Note, the portion of July-September typhoons could not be a majority for the whole typhoon season tropical cyclone genesis frequency from June to October.

Response: In this study, we focus on typhoons during July–September. We agree that the portion of July-September typhoons could not be a majority for the whole typhoon season from June to October. However, the meridional dipole in response to TNAO does not exist in all months of whole typhoon season, as shown in **Fig. R2.12**. For the whole season from June to October, the meridional dipole of typhoon track density linked to TNAO is weaker (**Fig. R2.12a** versus **b**). More specifically, the dipole with positive anomalies in the northern region and negative anomalies in the southern region only appears from July to September, while no significant dipole is observed in June, October, or November (**Fig. R2.12c-h**). Therefore, it is most appropriate to designate July to September as the primary seasonal period for TNAO's influence on TTD. We have incorporated this information into the method section:

Lines 391–394: “Although TNAO exerts some influence on typhoon activity in other months, its relationship with the TTD mode is mainly concentrated in the peak season (i.e., July-September) when WNP typhoon most active (Supplementary Fig. 22). Thus we mainly focus on July–September.”

Fig. R2.12 (i.e., Supplementary Fig. 22). Seasonal and monthly typhoon anomaly in response to TNAO. Correlation of typhoon track density in (a) June–October and (b) July–September. Regression of typhoon track density in (c) June, (d) July, (e) August, (f) September, (g) October and (h) November onto the spring TNAO index during 1979–2023. Rectangles are the same as in Fig. 2b.

Reviewer #3 (Remarks to the Author):

Summary: The manuscript identifies and characterizes a novel springtime North Atlantic seesaw—termed the Tropical-leaning North Atlantic Oscillation (TNAO)—and demonstrates its predictive linkage to summer typhoon-track density (TTD) over the western North Pacific via modulation of the Pacific Meridional Mode (PMM). By defining a spring sea level pressure (SLP) difference index, the authors show robust correlations ($R \approx 0.75$) with the leading empirical orthogonal function (EOF) of typhoon tracks, far exceeding the classical NAO's skill. They propose two cross-basin pathways: (1) TNAO-driven anticyclonic flow over the Northeastern Pacific weakens trades, warms SST, and seeds the PMM; and (2) an Atlantic SST tripole forces a Rossby response that reinforces the PMM. CMIP6 projections under SSP5-8.5 indicate increases in positive TNAO events, suggesting enhanced future typhoon poleward migration. Overall, the conceptual advance is substantial, linking two ocean basins through a physically plausible teleconnection with seasonal forecasting potential.

General evaluation: This study presents a new cross-basin teleconnection with strong predictive implications, making a substantial and well-validated contribution to our understanding of cross-basin climate–typhoon linkages and offering a seasonal forecasting potential. Pending clarifications on several issues described below and some extended analysis to strengthen the robustness, I recommend acceptance after a major revision. I believe that once these points are satisfactorily resolved, the paper will be well suited for Nature Communications.

Response: We sincerely appreciate the reviewer's thorough evaluation and encouraging remarks. We are grateful for the recognition of our study's conceptual advance in identifying a novel cross-basin teleconnection with strong predictive implications. We have carefully addressed all the concerns raised and have substantially revised the manuscript to enhance the robustness, clarity, and overall

presentation of our findings. Please check our response below.

Major Comments

1. Asian high-latitude cities: In the title, the authors emphasized the typhoon activity toward Asian high-latitude cities. However, throughout the manuscript, the reviewer could not find any analysis on landfalling typhoons that threatened cities in the relevant regions. The authors should explicitly demonstrate the point by showing the increase in landfall frequency. An analysis of landfall frequency or distance from coast would help clarify the claim.

Response: We thank the reviewer for this valuable comment. We agree that the societal significance of our work lies in the increasing threat of poleward-shifting typhoons to high-latitude Asian cities. In the original manuscript, we have quantified this threat using track density (representing the number of typhoons affecting each grid point) and Accumulated Cyclone Energy (ACE, which further accounts for wind intensity). These two metrics already capture the exposure and destructive potential of typhoons for specific cities (see Fig. 2c). Nevertheless, we acknowledge that these metrics may not be as intuitive as landfall frequency. Therefore, in the revised manuscript, we additionally provide the anomaly of typhoon frequency entering two representative boxes (see Fig. 2b). We revised related contents as follows:

Lines 163–173: “Moreover, as shown in Fig. 2c, positive TNAO can significantly predicts the above-normal **typhoon occurrence frequency (measured by track density) and destructive powers (measured by Accumulated Cyclone Energy³², ACE, see Methods) of typhoons** four months in advance, for East Asian high-latitude cities such as Tokyo, Osaka, Kagoshima, Seoul, Qingdao, and Shanghai; meanwhile, the below-normal destructive powers of typhoons for East Asian low-latitude cities like Hong Kong, Manila and Haikou can also be foreseen. **We further measure the typhoon frequency in each box in response to TNAO anomaly (see bar in Fig. 2b). A one standard deviation change in the TNAO index corresponds to an increase of 1.0 ± 0.45 in typhoon frequency in northern East Asia (see red box) and a**

decrease of 0.65 ± 0.45 in typhoon frequency in southern East Asia (see blue box).”

Fig. R3.1 (i.e., Fig. 2). Impacts of spring TNAO on summer Asia–Pacific typhoon activity. a Scatter diagram of the spring TNAO (red) and extratropical NAO (blue) against the TTD of peak season typhoon track density during 1979–2023. Correlation coefficient (R) between the TNAO (extratropical NAO) and TTD is marked. Straight Lines –are linear least-square fits. **b** Regressions of peak season typhoon track density onto the TNAO (shading) and TTD (contours) indices. Bars in the upper-right corner are the regression of typhoon frequency entering northern East Asia (NEA, red box) and southern East Asia (SEA, blue box) onto the TNAO index, with the error bar showing the 95% confidence level. **c** Regressions of peak season local track density (blue) and Accumulated Cyclone Energy (red) onto the spring TNAO index for Northeast and Southeast Asian regions (marked in **b** with red and blue boxes, respectively) and nine major cities (marked in **b**). The error bar is the 95% confidence

interval. Solid (open) circle indicates passing the 99% (95%) confidence level. Three extreme El Niño years (1982, 1997 and 2015) are excluded.

2. Long-term trend vs internal climate variability: Given global warming, both Atlantic and Pacific climates have a warming trend. Have the authors removed a linear (or higher-order) trend from their SLP/SST fields before computing TNAO and PMM? Note that a lingering trend could spuriously enhance their teleconnection statistics.

Response: The reviewer's perspective is well-founded, and it is crucial to investigate whether trend effects influence our findings. After removing trends through linear regression, we recalculated TNAO, TTD, and related atmospheric-ocean circulation anomalies. The results indicate that trends generally do not affect our conclusions (Figures R3.2–R3.3), with slightly changes observed in the relationship between TNAO and TTD or associated physical processes. As we aim to emphasize TNAO's potential role in typhoon northward movement and its possible impacts from future changes, the main text retains the trend-removed results, while Figure R3.2–R3.3 serves as a supplementary illustration. The manuscript has been revised as follows:

Lines 242–246: “We further examined the above results after removing the linear trend and found that the relationship between TNAO and TTD is still robust, and the related atmospheric and oceanic processes are also significant (Supplementary Figs. 15–16), indicating that the TNAO mode and its climatic influence are not a signal of global warming trend, but internal climate variability.”

Fig. R3.2 (i.e., Supplementary Fig. 15). TNAO modulations on typhoon and circulation. Same as Fig. 2a–b, 3a–c, but trend is removed. a Scatter diagram of the spring TNAO (red) and extratropical NAO (blue) against the TTD of peak season typhoon track density. Correlation coefficient (R) between the TNAO (extratropical NAO) and TTD is marked. Straight Lines –are linear least-square fits. **b** Regressions of peak season typhoon track density onto the TNAO (shading) and TTD (contours) indices. **c** Shading in the Pacific (Atlantic) is the correlation pattern of SST in July–September onto the simultaneous PMM index (April–June NAT index). StreamLines –are the regressed 850-hPa horizontal wind onto the PMM+NAT index (WNP region).

Contours indicate three atmospheric systems: monsoon trough (MT; represented by the relative vorticity of $3 \times 10^{-6} \text{ s}^{-1}$ at 850 hPa), tropical upper-troposphere trough (TUTT; represented by the 12420-gpm geopotential height at 200 hPa) and South Asian high (SAH; represented by the 12500-gpm geopotential height at 200 hPa). Solid (dashed) contours are calculated as the climatology plus (minus) the double regressions onto the PMM+NAT index. **d** Regressions of peak season Genesis Potential Index (shading) and typhoon genesis density (contour; 10^{-1} yr^{-1}) on the spring TNAO index. **e** Regressions of peak season 700-hPa geopotential height (shading) and steering flow (vector; $p < 0.05$) onto the spring TNAO index. Hatching indicates that it passes the 95% confidence level.

Fig. R3.3 (i.e., Supplementary Fig. 16). SST evolution related to TNAO. Same as Supplementary Fig. 5, but trend is removed. **a** Correlation pattern of SST in April–June (shading) and 850-hPa horizontal wind in March–May (vector) onto the spring TNAO index. **b** Lead-lag correlation of seasonal PMM onto the spring TNAO index.

The horizontal dashed line denotes the threshold of the 95% confidence level. **c–d** Same as **b** but for seasonal NAT and spring TNAO index, and seasonal PMM and April–June NAT index, respectively.

3. Memory from prior seasons: Spring TNAO events may be influenced by preceding winter’s modes (e.g., ENSO, IOD, PDO, AMO). Ensuring the predictor is not contaminated by autoregressive persistence will clarify the true lead-time benefits.

Response: The reviewer's suggestions are very important. Therefore, we present the correlation coefficients between these SST modes and TNAO and TTD for the previous winter. As shown in **Table R3.1**, none of these SST modes in preceding season exhibit significant linear correlations with either the TNAO or TTD. This indicates that the influence of TNAO on TTD operates independently of these SST modes, highlighting the unique predictive role of TNAO. We discussed this table to clarify the true lead-time benefits:

Lines 285–294: “In addition to the above climate variables, SST variability often has a significant influence on typhoons. However, unlike metrics such as typhoon genesis location, frequency or intensity⁴⁷⁻⁵², the relationship between other SST variability and TTD remains unexplored. Therefore, we further calculate the correlation coefficients between the winter-spring SST indices and the TNAO as well as the TTD. As shown in Supplementary Table 2, there was no significant linear correlation between TNAO (or TTD) and other existing indices (Niño3.4, PDO, AMO and Indian Ocean Dipole). This suggests that the influence of TNAO on the TTD is generally independent of these SST modes. These findings also highlight the unique predictive significance of the TNAO for the northward shift of East Asian typhoons.”

Table R3.1 (i.e., Supplementary Table 2). Relationship between preceding winter/spring modes and TNAO/TTD. Correlation coefficients of preceding winter (December–January–February, DJF) and spring (March–April–May, MAM) Niño3.4,

AMO, PDO, Indian Ocean Dipole (IOD) with spring TNAO and peak season (July–August–September, JAS) TTD during 1979–2023. None of these correlations is significant at the 90% confidence level of Student’s *t*-test.

Season of SST indices		Niño3.4	AMO	PDO	IOD
DJF	TNAO_MAM	−0.19	0.15	0.01	−0.16
	TTD_JAS	−0.12	0.03	0.00	−0.05
MAM	TNAO_MAM	−0.12	−0.18	0.19	−0.18
	TTD_JAS	−0.04	−0.13	0.11	−0.08

4. Pacific-basin wind-anomaly: The manuscript shows only the Atlantic (NAT/TNAO) 850 hPa anomaly map (Supplementary Fig. 5a), but omits the analogous Pacific anomaly field demonstrating trade-wind weakening. Please include a spatial map of 850 hPa wind anomalies regressed onto the PMM (or TNAO) over the Pacific to visually confirm the westerly anomalies in the 8–18° N band.

Response: Done as suggestion. Here we provided **Fig. R3.4–R3.5** to show the weakened trade winds over the subtropical eastern North Pacific (see rectangle in figures), and the anomalous 850 hPa winds. We confirm that there are westerly winds from the subtropical eastern North Pacific to the WNP that favors the weakened trade winds. We have revised the manuscript to give a more detailed description of the relevant physical processes:

Lines 176–189: “Mechanisms of TNAO performance in predicting TTD-like typhoon activity can be explained as follows. On the one hand, the TNAO can promote the Pacific Meridional Mode (PMM) SST anomalies through the anticyclonic flows over the northeastern Pacific (Supplementary Figs. 8a–b). The anticyclonic flows can weaken local trade winds and warm SST in the subtropical eastern North Pacific (red rectangle in Supplementary Fig. 9), which subsequently drives PMM development via the seasonal footprint mechanism^{27,35}. On the other hand, the TNAO can induces the NAT SST anomalies (Supplementary Fig. 8a). Specially, the

TNAO-related anomalous northeasterly winds over the tropical North Atlantic (TNA) can enhance the trade winds to cool SST there (blue rectangle in Supplementary Fig. 9). This cold SST can further strengthen the anticyclone over the eastern Pacific–North Atlantic via the Gill-type Rossby wave response^{36,37}. Therefore, this cold SST can persist into the following summer (Supplementary Fig. 8c) and further contribute to the development of PMM (Supplementary Fig. 8d) through a Gill-type Rossby wave response with an anticyclone to its west^{34,38-40}.

Fig. R3.4 (i.e., Supplementary Fig. 11). TNAO-associated atmospheric anomaly. a Regression of 850-hPa horizontal wind (vector) and correlation of precipitation (shading) during March–May onto TNAO index. Red rectangle is the same as in Supplementary Fig. 9. **b** Same as **a** but variables are during July–September. **c–d** Same as **a–b** but for 200-hPa horizontal wind and outgoing longwave radiation. All the shadings and vectors are significant at the 95% confidence level.

Fig. R3.5 (i.e., Supplementary Fig. 9). Climatology of SST and 850-hPa horizontal wind over the Pacific and the Atlantic during March–May 1979–2023. Red (blue) vectors indicate that climatic wind will enhance (weaken) associated with positive TNAO (diagnosed by climatic wind speed compared with wind speed plus the regression onto TNAO index). Black contour in the Pacific is the 0.5 correlation coefficient of SST onto the July–September PMM index, representing the origin of PMM. Red and blue rectangles show the regions of PMM development and NAT maintenance, respectively.

5. Confounding climate modes: Other low-frequency modes (AMO, PDO/IPO), ENSO, and IOD may influence both TNAO and typhoon metrics, potentially biasing the TNAO→TTD link. The authors should perform multivariate or partial-correlation tests that control for AMO, PDO/IPO, ENSO (Niño 3.4), and IOD indices and to confirm that TNAO retains unique predictive skill.

Response: The reviewer's point is valid. These low frequency modes are important for typhoons. In our response to Major comment 3, we find that these SST modes has little linear relationship with TNAO or TTD. Thus Table R3.1 confirm that TNAO retains unique predictive skill. Please check Major comment 3 for details. As shown in **Table R3.1**, none of these SST modes in preceding season exhibit significant linear correlations with either the TNAO or TTD. This indicates that the

influence of TNAO on TTD operates independently of these SST modes, highlighting the unique predictive role of TNAO.

Lines 285–294: “In addition to the above climate variables, SST variability often has a significant influence on typhoons. However, unlike metrics such as typhoon genesis location, frequency or intensity⁴⁷⁻⁵², the relationship between other SST variability and TTD remains unexplored. Therefore, we further calculate the correlation coefficients between the winter-spring SST indices and the TNAO as well as the TTD. As shown in Supplementary Table 2, there were no significant linear correlation between TNAO (or TTD) and other existing indices (Niño3.4, PDO, AMO and Indian Ocean Dipole). This suggests that the influence of TNAO on the TTD is generally independent of these SST modes. These findings also highlight the unique predictive significance of the TNAO for the northward shift of East Asian typhoons.”

Table R3.1 (i.e. Supplementary Table 2). Relationship between preceding winter/spring SST modes and TNAO/TTD. Correlation coefficients of preceding winter (December–January–February, DJF) and spring (March–April–May, MAM) Niño3.4, AMO, PDO, Indian Ocean Dipole (IOD) with spring TNAO and peak season (July–August–September, JAS) TTD during 1979–2023. None of these correlations is significant at the 90% confidence level of Student’s *t*-test.

Season of SST indices	R	Niño3.4	AMO	PDO	IOD
DJF	TNAO_MAM	−0.19	0.15	0.01	−0.16
	TTD_JAS	−0.12	0.03	0.00	−0.05
MAM	TNAO_MAM	−0.12	−0.18	0.19	−0.18
	TTD_JAS	−0.04	−0.13	0.11	−0.08

6. Potential confounding from monsoon and shear: Summer monsoon trough position

and vertical wind shear directly modulate both genesis and track of typhoons over the western North Pacific. Even if PMM explains part of the signal, the residual may simply be a shifted monsoon trough or the Pacific TUTTs. Could the authors provide any insights on possible connection? At least, this should be discussed briefly to demonstrate the PMM's unique pathway as discussed in the manuscript.

Response: To provide insights on the connection between the zonal shift of TUTT/MT and PMM/NAT, we first calculate the longitude of TUTT and MT (see caption of **Fig. R3.7**) and then plot the correlation pattern of SST onto the time series of TUTT and MT longitude. Correlation between indices are also marked. The locations of TUTT and MT are primarily associated with the central Pacific El Niño and tropical Atlantic SST (**Fig. R3.7**). However, areas where PMM- and NAT-correlated SST (contours) and shading partially overlap can be observed, resulting in significant zonal correlations ($p < 0.01$) between these regions and TUTT/MT. These findings confirm that TNAO-induced succeeding SST anomalies can indeed trigger zonal shifts in TUTT and MT locations. The article is revised as follows:

Lines 210–213: “Here we further confirm the relationship between TNAO-related SST anomalies and the zonal shift of TUTT and MT. The NAT and PMM SST patterns can partially contribute to the longitude changes in TUTT and MT since their correlations are significant (Supplementary Fig. 13).”

Supplementary Methods: “The TUTT longitude is defined as the western boundary of the zero contour of zonal wind speed over 5°N – 20°N ⁸⁴. The MT longitude is defined as the eastern boundary of $3 \times 10^{-6} \text{ s}^{-1}$ at 850 hPa over 0° – 20°N ⁸⁵.”

References:

Wu, L., Wang, C. & Wang, B. Westward shift of western North Pacific tropical cyclogenesis. *Geophys. Res. Lett.* 42, 1537-1542 (2015).

Zhang, C. et al. Perspective on landfalling frequency and genesis location variations of

southern China typhoon during peak summer. *Geophys. Res. Lett.* 46, 6830–6838 (2019).

Fig. R3.7 (i.e., Supplementary Fig. 13). Relationship of zonal shift in TUTT and MT with SST during July–September. a Time series of TUTT (blue) and MT (red) longitude during 1979–2023. **b** Correlation of SST onto TUTT index. Hatching indicates correlation pass the 95% confidence level. Red (blue) contour is the 0.6 (0.3) correlation of SST onto the July–September PMM index (April–June NAT index). Correlation coefficients between TUTT and PMM/NAT are marked. **c** Same as **b** but for MT index.

7. Regression-training sample size and robustness: Training the TNAO→TTD regression on 1979–2014 (36 yr) risks sampling uncertainty; the validation period (2016–2023) showing higher R^2 may reflect lucky sampling or non-stationarity. The authors can conduct sensitivity tests (e.g., leave-one-decade-out, bootstrap/jackknife) to quantify confidence intervals and show windowed or running regressions over 1979–2023 to assess temporal stability of the linkage.

Response: Done as suggestion. We conducted bootstrap test and running regressions as shown in **Fig. R3.8**. Bootstrap test confirms the significant role of TNAO in modulating TTD, while running regression confirms the stability of TNAO-TTD relationship. We revised the manuscript as follows:

Lines 263–270: “To further validate the stability of the TNAO-TTD relationship and the prominent importance of TNAO relative to other factors, we employed the bootstrap method to assess the error range of regression coefficients and used the sliding regression to test the temporal stability of coefficients (Supplementary Fig. 17). This prediction is generally independent of ENSO (Supplementary Fig. 18). Results indicate that only the TNAO coefficient in the regression model is statistically significant. Although the relationship between TNAO and TTD exhibits some fluctuation, it remains statistically significant throughout the study period.”

And the method is as follows:

Supplementary Methods: “To estimate the uncertainty of the regression coefficients and correlation for multiple linear regression, a nonparametric bootstrap resampling method⁸⁶ was applied to the data during 1979–2023 (all years included). For each of 5000 bootstrap iterations, annual samples (corresponding to 45 years) were randomly resampled with replacement from the original dataset to form a pseudo-sample of equal size. Data are allowed to be selected repeatedly. Multiple linear regression was then performed on each resampled dataset to obtain a new set of regression coefficients and the corresponding correlation between the predicted and observed values. The distribution of these 5000 estimates was used to compute the 95% confidence intervals, defined by the 2.5th and 97.5th percentiles.”

Reference:

Austin, P. C. & Tu, J. V. Bootstrap methods for developing predictive models. *The American Statistician* 58, 131-137 (2004).

Fig. R3.8 (i.e., Supplementary Fig. 17). Stability of TNAO-TTD relationship. a Bootstrap estimation (Supplementary Methods) of regression coefficients and correlation coefficient for predicting TTD using five climate indices (TNAO, NAO, NAM, NPO, SIC) and based on data during all years from 1979 to 2023. Blue bars indicate the mean regression coefficients, with black error bars representing the 95% confidence interval. The red bar shows the mean correlation between observed and predicted TTD with its 95% confidence interval. **b** Sliding regression between TNAO and TTD over a 19-year moving window. The blue line with markers shows the regression in each window, and the shading indicates the 95% confidence interval.

8. Mechanistic quantification: The proposed pathways (thermodynamic PMM vs. direct circulation steering) are described qualitatively. Could the authors provide a quantitative analysis to show the relative importance of thermodynamic and dynamic factors in the GPI analysis and quantify the consistent steering change? The authors may also include precipitation/OLR composites to verify convective amplification

under the PMM.

Response: Done as suggestion. For the GPI quantification, we follow the following methods:

Lines 445–452: “To attribute GPI anomalies associated with the TNAO to individual environmental factors, a linear decomposition was applied⁷⁸. For each environmental factor, the corresponding TNAO-related anomaly was substituted into the GPI calculation, while all other variables were retained at their climatological states. The total GPI anomaly was subsequently approximated as the sum of the contributions from all factors. This approach enables the relative roles of vertical wind shear (VWS), low-level relative vorticity (Vor), mid-level vertical velocity (Omega), mid-level relative humidity (RH), and potential intensity (Vpot) in modulating TC genesis potential to be quantitatively assessed.”

And a detailed analysis for each variable is given, as well as the quantification for steering flow changes (**Figs. R3.9-R3.10**):

Lines 221–233: “To show the relative importance of thermodynamic and dynamic factors in the GPI analysis, a linear decomposition is applied (see Methods). The contributions of each variable are calculated, then the regional-mean is obtained for eastern and western regions (outlined in Fig. 3b) associated with positive and negative GPI anomalies, respectively (Supplementary Fig. 13). The increased GPI in the eastern WNP is mainly contributed by increase mid-level relative humidity, followed by increased lower-level vorticity, ascending motion and increased potential intensity (Fig. 3d). While the vertical wind shear does not significantly contribute to GPI changes. In contrast, the decreased GPI in the western WNP is dominantly contributed by increased vertical wind shear (Fig. 3e). We also quantified the zonal steering flow changes (Fig. 3f), given that the meridional component of steering flow change is negligible. Both northern and southern regions (outlined in Fig. 3c) show significant change, suggesting important guiding effects for typhoon movement.”

Fig. R3.9 (i.e., Supplementary Fig. 14). Diagnosis of the GPI associated with the springtime TNAO. a–f Regression of peak season GPI **a** and its individual contributing terms during 1979–2023: **b** vertical wind shear (VWS), **c** absolute vorticity (Vor), vertical velocity **d** (Omega), **e** relative humidity (RH), and **f** maximum potential intensity (Vpot) onto the TNAO index during spring. Black boxes denote the eastern (137.5°–170°E, 10°–27.5°N) and western (110°–135°E, 7.5°–17.5°N) analysis regions.

Fig. R3.10 (i.e., Fig. 3) TNAO modulations on typhoon genesis and track. TNAO modulations on typhoon genesis and track. **a** Shading in the Pacific (Atlantic) is the correlation pattern of July–September SST onto the simultaneous PMM index (April–June NAT index) during 1979–2023. Streamlines are the regressed 850-hPa horizontal wind onto the PMM+NAT index (WNP region). Three atmospheric systems are shown: monsoon trough (MT; represented by the relative vorticity of $3 \times 10^{-6} \text{ s}^{-1}$ at 850 hPa), tropical upper-troposphere trough (TUTT; represented by the 12420-gpm geopotential height at 200 hPa) and South Asian high (SAH; represented by the 12500-gpm geopotential height at 200 hPa). Solid (dashed) contours are calculated as the climatology plus (minus) the double regressions onto the PMM+NAT index.

Green contours are the correlation of OLR onto PMM+NAT index, with interval same as the color bar (solid for positive and dashed for negative). **b** Regressions of peak season Genesis Potential Index (shading) and typhoon genesis density (contour; 10^{-1} yr^{-1}) on the spring TNAO index. **c** Regressions of peak season 700-hPa geopotential height (shading) and steering flow (vector; $p < 0.05$) onto the spring TNAO index. Hatching indicates that it passes the 95% confidence level. **d–e** Regional mean GPI diagnosis in the **d** Box 1 and **e** Box 2 (outlined in **b**). Bars show the GPI anomalies and contributions from each term, with error bars indicating the 95% confidence intervals. “Sum” denotes the sum of VWS term, Vor term, Omega term, RH term and Vpot term. **f** Zonal steering flow anomalies regressed onto TNAO in Box 3 and Box 4 in (outlined in **c**).

9. Operational predictability: To underscore novelty, showing how a simple TNAO-based statistical forecast compares against state-of-the-art dynamical seasonal typhoon outlooks (e.g., ECMWF, GFS) in hindcast mode would demonstrate added value of the finding. Does the regression model outperforms the existing operational models?

Response: We thank the reviewer for this insightful suggestion. We fully agree that comparing the TNAO-based statistical forecast with state-of-the-art dynamical seasonal typhoon outlooks (e.g., ECMWF SEAS5, CFSv2) would be valuable in further highlighting the novelty and potential added value of our approach. However, at present such a comparison is not feasible, because the operational seasonal forecasting centers only provide limited tropical cyclone products in graphical form (**Fig. R3.11**), without downloadable data fields. In particular, the ECMWF SEAS5 “tropical storm density anomaly” is only available as charts. While the CFSv2 system provides general atmospheric and oceanic variables but does not directly provide tropical cyclone density or track products. A rigorous comparison would thus require re-processing the raw seasonal forecast fields using dedicated cyclone tracking algorithms, which is beyond the scope of this study. We further note that the TTD

mode is not a metric similar to tropical cyclone genesis or landfall frequency, but a leading mode that needs to be identified from track density. Due to the current difficulties in data acquisition and calculation, we regret that we cannot compare the regression model with the existing operational models.

Nevertheless, we emphasize that the main objective of our work is to demonstrate the physical role of the newly identified TNAO in modulating typhoon activity, and to illustrate its potential as a simple statistical predictor compared with other **major climate modes** such as ENSO, AMO, PDO, NPO and NAO. We added discussions to highlight the possible value of TNAO research:

Lines 295–299: “Above results demonstrate that the TNAO provides physically grounded and statistically independent information for predicting typhoon track variability, enriching and deepening our understanding in East Asian typhoon activity besides other major climate modes. Future work may focus on TNAO influence in operational dynamical forecast models to compare the ability of TNAO in forecasting typhoon track pattern.”

Besides, we have further added discussion on Relationship between TNAO and WNP cyclone in CMIP6 models (Fig. R3.13, i.e., Supplementary Fig. 20). Please see the **Response** to the following **Comment #11** for details.

a

Tropical cyclone tracks - high-resolution products

Long name	Format	Type of level	Steps for times 00 &12	Steps for times 06 &18
Tropical Cyclone Trajectory (TC track including genesis)	BUFR	sfc	up to step 240	up to step 90

b

Fig. R3.11. Tropical cyclone forecast data in ECMWF. a Description of tropical cyclone trajectory data in ECMWF forecast data. Screenshot from <https://www.ecmwf.int/en/forecasts/datasets/open-data> . **b** Tropical storm standardised density production in SEAS5. Screenshot from https://charts.ecmwf.int/products/seasonal_system5_tstorm_density_anomaly . Screenshot was obtained by Oct 1, 2025.

10. Physical process diagrams or schematic: Finally, a concise schematic that lays out both the direct anticyclone-driven trade weakening and the Atlantic-tripole/Gill-mode response pathway would help readers keep track of the multiple legs of the cross-basin teleconnection.

Response: Done as suggestion. Here we provided a schematic diagram (Fig. R3.12) at the end of the summary paragraph, and the detailed description is given in its caption:

Line 370: “A schematic diagram is presented in Fig. 6.”

Fig. R3.12 (i.e., Fig. 6). Schematic diagram illustrating the influence of TNAO on WNP typhoons. During spring, the TNAO weakens trade winds over the subtropical northeastern Pacific, leading to local SST warming and subsequent development of the PMM through the seasonal footprint mechanism. Concurrently, TNAO strengthens

the trade winds over the TNA, resulting in SST cooling that persists into summer via local air-sea coupling. The cold TNA SST anomalies can continuously induce Rossby wave anticyclone to its west, thereby further promoting PMM development. The Atlantic-eastern Pacific Rossby wave anticyclone has been demonstrated to directly favor the WNP cyclone via Pacific convection, a phenomenon that is explained by the PASO. Consequently, the presence of a lower-level cyclone and an upper-level anticyclone in the WNP leads to an eastward shift in the TUTT and MT (solid line for positive TNAO and dashed line for negative TNAO), promoting more (less) frequent typhoon genesis in the eastern (western) WNP. The altered steering flow favors (inhibits) typhoon movement towards higher (lower) latitudes of East Asia.

11. Translation from projected TNAO to northward shift of typhoon activity: The authors show an increased frequency of positive TNAO events under SSP5-8.5, but how that translates into societal risk (e.g. changes in ACE, landfall probability, economic exposure) remains unexplored. A brief discussion would help readers grasp the real-world implications. It will be much convincing if the authors can demonstrate the connection by analyzing typhoon activity in CMIP6 models ensemble simulations.

Response: We thank reviewer's insightful comments. We added some discussions as follows to show how projected TNAO change may translate into societal risk. We acknowledge that it will be much convincing if we demonstrate the typhoon activity in CMIP6 models ensemble simulations. However, the CMIP6 outputs we used (see Methods) are monthly data and have a coarse resolution, with most grid cells spaced approximately 200-km apart. Therefore, we regret that we are unable to carry out this work at present.

Nevertheless, we investigated the future changes in steering flows over the WNP in response to TNAO. As shown in **Fig. R3.13**, climate models can well capture the strengthening cross-seasonal linkages between the WNP anomalous cyclone and preceding spring TNAO with an intensification as the climate warms. Moreover, the TNAO mode obtained from the CMIP6-MME well mirror the observed one (Fig. 5b

vs Fig. 1b; with a robust pattern correlation, $R = 0.97$). This result indirectly lends more confidence to our hypothesis. We added related discussion as follows:

Lines 306–325: “As shown in Fig. 5a, the CMIP6 models generally (23 out of 26 models) project an increase (~39.4%) in positive TNAO events during the 21st century (23.96 ± 3.28 events per 100 years), compared to that during the 20th century (17.19 ± 1.85 events per 100 years). Moreover, the TNAO mode obtained from the CMIP6-MME well mirror the observed one (Fig. 5b vs Fig. 1b; with a robust pattern correlation, $R = 0.97$).

The increase in positive TNAO event is most pronounced toward the end of the 21st century, exceeding 20 events per 100 years (Fig. 5c). Moreover, there are 7 out of 26 CMIP6 models can even capture the strengthening cross-seasonal linkage between the WNP anomalous cyclone and preceding spring TNAO (Supplementary Fig. 20), further implying more prominent impacts on typhoon track potentially from TNAO as the climate warms. In addition, the future anthropogenic warming would decelerate typhoon motions near high-latitude populated coastal regions due to a poleward shift of the subtropical westerlies, which can induce a longer time of typhoon impacts⁵⁷. And accompanying with increasing intensity and rainfall⁵⁸, this can potentially compounds future typhoon-related damages. Since based on our current results, if the increase in positive events of TNAO can cause the northward shift of typhoon track, high-latitude cities in East Asia would face more typhoon frequency, greater ACE and increased economic losses in future. Therefore, we have every reason to infer that the poleward migration of typhoon activity induced by TNAO in a warming future would exacerbate the societal risk from typhoons in East Asian high-latitude cities as the century progresses.”

Fig. R3.13 (i.e., Supplementary Fig. 20). Relationship between TNAO and WNP cyclone in CMIP6 models. a Correlation coefficient between spring TNAO and peak season cyclonic steering flow in the WNP during 1900–1999 (blue bars) and 2000–2099 (red bars). The selected 7 ensembles are in the shaded background with error bar showing the 95% confidence interval for the selected MME. The steering flow index is defined as zonal steering flow difference between (12.5°N–22.5°N, 130°E–170°E) and (27.5°N–37.5°N, 115°E–155°E). **b** Mean of regression patterns of peak season steering flow (vector) onto the spring TNAO index during 1900–1999 for the selected 7 ensembles. Shading is the correlation of zonal component of steering flow onto the TNAO index. **c** Same as **b** but during 2000–2099.

Minor Comments

1. The global analyses (EOFs, regressions, composites) lack a clear statement of their full temporal coverage (e.g., 1979–2023) beyond the training/validation split. Therefore, please explicitly state the data period in the Methods and all Fig. captions

(e.g., “analyses based on 1979–2023”).

Response: Done as suggestion. Data period was stated the Methods and all Fig./Table captions, including:

“**Fig. 1. Spatial pattern of TNAO and its dynamical features.** a Correlation between zonally-averaged SLP anomalies in 25°W–105°W during 1979–2023.”

“**Fig. 2. Impacts of spring TNAO on summer Asia–Pacific typhoon activity.** a Scatter diagram of the spring TNAO (red) and extratropical NAO (blue) against the TTD of peak season typhoon track density during 1979–2023.”

“**Fig. 3. TNAO modulations on typhoon genesis and track.** a Shading in the Pacific (Atlantic) is the correlation pattern of July–September SST onto the simultaneous PMM index (April–June NAT index) during 1979–2023.”

Methods: “We focus on the East Asia–WNP typhoon (TC with $V_{\max} > 32.6 \text{ m s}^{-1}$) during July–September of 1979–2023, using the Joint Typhoon Warning Center TC best-track datasets.”

2. “high summer” vs. “peak season”: The manuscript alternates between “high summer” (July–August) and “peak season” (July–September), which can confuse readers. Please keep “peak season” since all analyses seemed to have focused on peak season.

Response: Done as suggestion. We have replaced all “high summer” with “peak season”.

3. Please give the rationale for excluding extreme El Niño years from the analysis and briefly discuss and explicitly show its potential impact.

Response: Examining the potential influence of ENSO is important. Therefore, we provided scatter plots between Niño3.4, TNAO, and TTD under different conditions (**Fig. R3.13**). During 1979–2023, Niño3.4 and TNAO show a weak correlation ($R = 0.28$, $p = 0.07$; **Fig. R3.13a**). After removing three strong El Niño

years, the correlation becomes insignificant ($R = 0.18$, $p = 0.25$; **Fig. R3.13b**). When we instead remove the three strongest La Niña years (1988, 1999, and 2007), the correlation also becomes insignificant (**Fig. R3.13c**), and the TNAO–TTD relationship ($R = 0.73$; **Fig. R3.13d**) slightly weakens compared to that in the main text ($R = 0.75$). Furthermore, excluding additional ENSO years yields a similar result ($R = 0.72$; **Fig. R3.13e**), again slightly lower than 0.75. These results confirm that our main conclusion is robust, ensuring that the TNAO–TTD correlation remains as high as possible while maintaining an insignificant linear relationship between TNAO and Niño3.4.

Lines 149–153: “Given that relationship between TNAO and ENSO is not significant (even in a longer period of 1958–2023; Supplementary Figs. 6a–e), we have tested and found that even in the case of excluding all ENSO years, the TNAO-TTD relationship still keeps robust ($R = 0.72$, $p < 0.01$; Supplementary Figs. 6f–g). Accordingly, the cross-seasonal TNAO-TTD linkages are generally independent of ENSO signals.”

Fig. R3.14 (i.e., Supplementary Fig. 6). Relationship between ENSO and TNAO.

a Scatter plot of spring TNAO versus preceding winter Niño3.4 index during 1958–2023. **b** Same as **a** but for peak season Niño3.4 index. **c** Same as **b** but during 1979–2023. **d** Same as **c** but excluding years 1982, 1997 and 2015. **e** Same as **d** but further excluding extreme La Niña years of 1988, 1999 and 2010 (selected by three lowest Niño3.4 index years). **f** Same as **e** but for TNAO versus TTD. **g** Scatter plot of spring TNAO versus TTD excluding El Niño and La Niña years that Niño3.4 index exceeding 1.0 absolute value. Correlation coefficients and their p-values are marked.

4. L146–151: Clarify that the anticyclonic flow over the northeast Pacific produces westerly anomalies on its equatorward flank, vectorially opposing the northeasterly trades since this could not be seen from Supplementary Fig. 5a.

Response: As shown in **Fig. R3.4 (i.e., Supplementary Fig. 11)**, the TNAO-related anticyclone can produce southwesterly anomalies over the subtropical northeastern Pacific, therefore weakening trade winds (Fig. R3.5, **i.e., Supplementary Fig. 9**) and contributing to the development of PMM. Please check our response to major comment 4 for details.

5. Lines 50–51 “While the impacts of shifting internal natural climatic oscillations on typhoon activity are often overlooked” is not true. There have been many studies that investigated the impacts of various internal long-term climate modes on typhoon activity, including PDO, IPO, AMO, etc. The authors should mention those here.

Response: Of note is that here we use a “shifting” to emphasize the shifting atmospheric oscillations. So our intention is to suggest that research of atmospheric oscillations impacts on typhoon is relatively less than other researches, shifting ENSO for example, on typhoon changes. Here we also added some studies on the impacts of various internal long-term climate modes (e.g., PDO, IPO, AMO) on typhoon activity. The manuscript is revised as follows:

Lines 49–53: “Impacts of North Pacific and Atlantic climate shifts on typhoon activity are also highlighted. For example, interdecadal changes in Northwest Pacific (WNP) tropical cyclone activity can be attributed to the Atlantic Multidecadal Oscillation (AMO), Interdecadal Pacific Oscillation (IPO) and/or Pacific Decadal Oscillation (PDO)¹⁴⁻²⁰. While the impacts of shifting internal atmospheric oscillations on changes in typhoon activity are often overlooked.”

References (14-20):

Cao, J., Feng, J., Zhao, H., Wang, B., Wu, L., & Wang, C. (2025). Comparable impacts of IPO and AMO on the decadal change of northern hemisphere tropical cyclone frequency. *Clim. Dyn.*, 63(1), 23. <https://doi.org/10.1175/JCLI-D-24-0731.1>

Chan, J. C., & Liu, K. S. (2022). Recent decrease in the difference in tropical cyclone

occurrence between the Atlantic and the western North Pacific. *Adv. Atmos. Sci.*, 39(9), 1387-1397. <https://doi.org/10.1007/s00376-022-1309-x>

Han, Y., Jiang, W., Jiang, L., Yong, Y., Yang, K., & Gu, T., et al (2024). Corals reveal interdecadal variation of tropical cyclones modulated by Pacific Decadal Oscillation. *J. Geophys. Res.-Atmos.*, 129(7), e2024JD040824. <https://doi.org/10.1029/2024JD040824>

Wang, C., Wang, B., Wu, L., & Luo, J. J. (2022). A seesaw variability in tropical cyclone genesis between the western North Pacific and the North Atlantic shaped by Atlantic multidecadal variability. *Journal of Climate*, 35(8), 2479-2489. <https://doi.org/10.1175/JCLI-D-21-0529.1>

Zhao, J., Zhan, R., Kim, D., Kug, J. S., Long, J., & Zhang, L., et al (2024). Distinct modulations of Northwest Pacific tropical cyclone precipitation by Atlantic Multidecadal Oscillation and Interdecadal Pacific Oscillation. *Geophys. Res. Lett.*, 51(12), e2023GL107749. <https://doi.org/10.1029/2023GL107749>

Zhao, J., Zhan, R., Wang, Y., Xie, S., & Wu, Q. (2020). Untangling impacts of global warming and Interdecadal Pacific Oscillation on long-term variability of North Pacific tropical cyclone track density. *Sci. Adv.*, 6(41), eaba6813. <https://doi.org/10.1126/sciadv.aba6813>

Zhao, J., Zhan, R., Wang, Y., & Xu, H. (2018). Contribution of the interdecadal Pacific oscillation to the recent abrupt decrease in tropical cyclone genesis frequency over the western North Pacific since 1998. *J. Clim.*, 31(20), 8211-8224. <https://doi.org/10.1175/JCLI-D-18-0202.1>

6. Although references are comprehensive, some recent work on cross-basin teleconnections where relevant should be included.

Response: We agree that there should be references on cross-basin teleconnections since we discuss a lot of relevant mechanisms. Therefore, the

following references are included:

References:

- Capotondi, A., Mcgregor, S., Mcphaden, M. J., Cravatte, S., Holbrook, N. J., & Imada, Y., et al (2023). Mechanisms of tropical Pacific decadal variability. *Nat. Rev. Earth Environ.*, 4(11), 754-769. <https://doi.org/10.1038/s43017-023-00486-x>
- Fu, M., Wang, C., Wang, B., Wu, L., Cao, J., & Zhao, H. (2025). Pacific Meridional Mode as a Bridge in Linking North Atlantic Oscillation and Interdecadal Variability in Tropical Cyclone Genesis in the Western North Pacific. *J. Clim.*, 38(19), 5431-5442. <https://doi.org/10.1175/JCLI-D-24-0731.1>
- Kao, P., Hong, C., Huang, A., & Chang, C. (2022). Intensification of interannual cross-basin SST interaction between the North Atlantic tripole and Pacific meridional mode since the 1990s. *J. Clim.*, 35(18), 5967-5979. <https://doi.org/10.1175/JCLI-D-21-0594.1>
- Li, S., Xiao, Z., & Zhao, Y. (2023). Modulation of the influence of ENSO on northward-moving tropical cyclones in the Western North Pacific by the North Atlantic tripole SST anomaly pattern. *J. Clim.*, 36(2), 405-420. <https://doi.org/10.1175/JCLI-D-21-0704.1>
- Park, J., Yeh, S., Kug, J., Yang, Y., Jo, H., & Kim, H., et al (2023). Two regimes of inter-basin interactions between the Atlantic and Pacific Oceans on interannual timescales. *npj Clim. Atmos. Sci.*, 6(1), 13. <https://doi.org/10.1038/s41612-023-00332-3>
- Tian, Q., Yu, J., Nnamchi, H. C., Li, T., Li, J., & Li, X., et al (2025). Unraveling the mystery of recent shortened response time of ENSO to Atlantic forcing. *Nat. Commun.*, 16(1), 5884. <https://doi.org/10.1038/s41467-025-61130-4>
- Zhu, Y., Xu, S., Huang, F., & Ding, Z. (2025). Modulation of landfalling tropical cyclone activity in the western North Pacific by North Atlantic tripolar SST. *Atmos. Res.*, 322, 108130. <https://doi.org/10.1016/j.atmosres.2025.108130>
- Zhang, L., Yang, X. & Zhao, J. Impact of the spring North Atlantic oscillation on the Northern Hemisphere tropical cyclone genesis frequency. *Front. Earth Sci.* **10**, 829791 (2022).
- Zhao, J., Zhan, R., Kim, D., Kug, J. S., Long, J., & Zhang, L., et al (2024). Distinct modulations of Northwest Pacific tropical cyclone precipitation by Atlantic Multidecadal Oscillation and Interdecadal Pacific Oscillation. *Geophys. Res. Lett.*, 51(12), e2023GL107749. <https://doi.org/10.1029/2023GL107749>

6. Line 58, remove “attribution of”.

Response: Corrected.

8. Line 537, change “exerts” to “exert”.

Response: Corrected.

Response to Reviewers' comments

REVIEWERS' COMMENTS

Reviewer #1 (Remarks to the Author):

I believe that the authors have addressed all my concerns. I am satisfied with current revision and suggest to accept this manuscript.

Reviewer #2 (Remarks to the Author):

It seems that all my concerns have been addressed in the revised MS and responses to reviewers. The reviewer felt that the MS could be accepted for publication.

Reviewer #2 (Remarks on code availability):

These codes are relatively complete and can be easily read and reproduced.

Reviewer #3 (Remarks to the Author):

Summary and Overall Assessment

The revised manuscript presents the identification of a previously unrecognized climate mode—the tropical-leaning North Atlantic Oscillation (TNAO)—and argues that this springtime SLP dipole is a strong and physically meaningful predictor of peak-season western North Pacific (WNP) typhoon track variability, particularly the poleward migration toward high-latitude East Asian cities. The authors propose two physical pathways linking TNAO to the Pacific Meridional Mode (PMM), tropical North Atlantic SST anomalies, and subsequent summertime steering flow anomalies. They also examine CMIP6 projections, suggesting more frequent positive TNAO events under SSP5–8.5.

This work proposes a novel and interesting cross-basin teleconnection with potential implications for seasonal typhoon forecasting. The revisions substantially improve the clarity, robustness, and physical justification of the results. The authors have responded carefully and extensively to reviewer comments, adding new analyses (trend-removed fields, ENSO controls, GPI decomposition, jet shift index revision, additional figures, sensitivity tests) and clarifying mechanisms.

Overall, the revised manuscript is strong and now suitable for publication after addressing a few remaining issues that concern clarity, over-interpretation, and the need for more cautious framing in several places as summarized in my comments given below.

Response: We thank the reviewer for the positive and constructive comments. We have addressed the remaining concerns regarding clarity, cautious interpretation, and framing throughout the manuscript. Please check our point-to-point responses below.

Remaining Issues to Address Before Acceptance

1. Over-extrapolation of CMIP6 TNAO projections to future typhoon risks: While the authors acknowledge limitations, some statements still imply direct future increases in typhoon hazards due to TNAO changes (e.g., Lines 324–329). CMIP6 outputs used here do not simulate typhoons directly, and steering flow differences alone cannot fully translate to future track density changes because the latter are also strongly affected by shifts in genesis locations. Furthermore, the response of TC activity over the WNP to the projected changes in TNAO might be also sensitive to the global warming-induced SST patterns in the Tropics (see a review by Wang et al. 2025), which has not been examined in this study and thus be mentioned in the discussion section. I thus suggest further soften any direct implication that increased positive TNAO would cause increased typhoon frequency or ACE near high-latitude cities and emphasize instead where applicable (including the abstract)“...may potentially contribute to...” “suggests possible increases in steering-flow patterns conducive to poleward migration...”.

Wang, Y., M. Satoh, R.-F. Zhan, J.-W. Zhao, and S.-P. Xie, 2025: Tropical sea surface warming patterns and tropical cyclone activity – A review, *Adv. Atmos. Sci.*, 42(10), 1996–2017, <https://doi.org/10.1007/s00376-025-5114-1>.

Response: We thank the reviewer for pointing out this important comment. We fully agree that in our study, CMIP6 models do not explicitly simulate typhoon directly and that steering-flow anomalies alone cannot be directly translated into

future changes in typhoon track density. We also agree that the WNP typhoon response to projected TNAO changes may depend on global-warming-induced tropical SST patterns (Wang et al. 2025), which has been mentioned in the revised discussion. Per the reviewer's suggestion, we have softened all statements that previously implied direct causal increases in future typhoon hazards. Specifically:

Lines 38–40: “This may potentially contribute to a poleward migration of typhoon activity/threats toward East Asian high-latitude cities as climate warms, yet uncertainty remains due to model biases in simulating tropical sea surface temperature patterns.”

Lines 332–336: “Based on our current results, the projected increase in positive events of TNAO may potentially contribute to increases in steering-flow patterns conducive to poleward migration of typhoon tracks. Such circulation changes imply the possibility that high-latitude cities in East Asia would face more typhoon threats and relevant economic losses in future.”

Lines 344–351: “Besides, CMIP6 outputs used here do not simulate typhoon tracks directly, and steering-flow changes alone cannot fully translate to future track density changes because the latter are also strongly affected by shifts in TC genesis location. Furthermore, considering that the influence of TNAO on WNP circulations depends on teleconnection processes between the ocean and atmosphere, the response of typhoon activity over the WNP to the projected changes in TNAO might be sensitive to the global warming-induced SST patterns in the tropics⁶³, thereby altering the TNAO-TTD relationship, also causing uncertainty.”

2. Clarify the added “city-box” frequency results: The added frequency increments (+1.0/–0.65) are useful but need more explanation, such as whether they are counts per season, or normalized by area, or if the two boxes are comparable in size, and the numbers should be area-adjusted. I suggest add a sentence explaining the units and comparability.

Response: We appreciate the reviewer's suggestion. In the revised manuscript,

we explicitly explain that the reported values (+1.0/ - 0.65) represent the change in the number of typhoons entering each box per peak season in response to a one-standard deviation change in the TNAO index. We also clarify that the two boxes differ in size and therefore provide an area-normalized comparison. Specifically, we have added the following sentences:

Lines 178–184: “The typhoon frequency is calculated by counting typhoon in peak season entering the boxes in Fig. 2b. A one standard deviation change in the TNAO index corresponds to an increase of $1.0 \pm 0.45 \text{ yr}^{-1}$ in typhoon frequency in northern East Asia (see red box) and a decrease of $0.65 \pm 0.45 \text{ yr}^{-1}$ in typhoon frequency in southern East Asia (see blue box). When normalized by area, the typhoon-frequency changes are similar in magnitude between the two regions, with about 0.46 yr^{-1} and 0.41 yr^{-1} typhoons per million square kilometers in the southern and northern boxes, respectively.”

3. Potential circularity between PMM definition and TNAO-forced Pacific SST: The PMM region used in the lead-lag analysis partly overlaps with the area of westerly anomalies produced by TNAO. While this is physically plausible, it can also appear self-reinforcing. I would recommend add 1–2 sentences acknowledging this potential dependence and explaining why the interpretation remains robust.

Response: Of note is that perhaps someone might think that the PMM region used in the lead-lag analysis (Supplementary Fig. 8) partly overlaps with the area of anticyclonic flows induced by TNAO (Supplementary Figs. 8a, 11a). However, this overlap does not imply a potential circularity between TNAO (or TNAO-forced Pacific SST) and PMM, because the summertime PMM index exhibits a clear lagged response to springtime TNAO index (Supplementary Fig. 8b). In other words, the summertime PMM is just as one of bridges linking TNAO signal to subsequent typhoon activity. To address this concern, we have rewritten relevant statements as follows:

Lines 188–193: “On the one hand, the springtime TNAO can directly promote the subsequent Pacific Meridional Mode (PMM) SST anomalies through the

TNAO-related anticyclonic flows (Supplementary Figs. 8a–b). Since this anticyclonic flows can weaken local trade winds in the subtropical Northeast Pacific (red rectangle in Supplementary Figs. 9 and 11a), and thereby the resultant warm SST (wind–evaporation–SST feedback mechanism) facilitates PMM’s development via the seasonal footprint mechanism^{27,35}.”

4. Strength of the jet-TNAO relationship: The NAJS–TNAO correlation ($R \approx 0.85$) is very high, implying the two indices may be nearly redundant. The manuscript should clarify whether TNAO provides additional information beyond NAJS for typhoon prediction and why TNAO is more directly connected to steering-flow anomalies than NAJS alone. A brief statement would resolve this.

Response: We appreciate the reviewer’s observation regarding the strong correlation between NAJS and TNAO. To clarify their distinction and relative roles, we have added an explanatory sentence in the revised manuscript:

Lines 165–171: “Of note is that NAJS exhibits a weaker correlation with TTD ($R = 0.64$), albeit NAJS and TNAO are highly correlated ($R = 0.85$; Supplementary Fig. 3c).. This may be due to that the NAJS index is defined from U200 anomalies within a smaller region of the upper-tropospheric jet stream. In contrast, TNAO is a lower-tropospheric SLP mode that spans a broader tropical domain, the associated air-sea interactions are inevitably stronger than that for NAJS, leading to a closer relationship from TNAO to TTD.”

Minor Comments

1. Lines 29–32, “covariance above 56%” — specify this refers to TNAO explaining 56% of TTD variance.

Response: Revised as suggestion.

2. Line 85, change to “...can directly capture the previously overlooked TNAO signal.”

Response: Revised as suggestion. **Line 85-86.**

3. “TUTT” appears as “tropical upper-tropospheric trough” in line 211 while “tropical upper-troposphere trough” in line 759. Use uniform terminology.

Response: Thanks. Done as suggestion.